# Training-Free Efficient Video Generation via Dynamic Token Carving

**Yuechen Zhang**[1] **Jinbo Xing**[1] **Bin Xia**[1] **Shaoteng Liu**[1] **Bohao Peng**[1]
**Xin Tao**[3] **Pengfei Wan**[3] **Eric Lo**[1] **Jiaya Jia**[2, 4]

[1]CUHK [2]HKUST [3]Kuaishou Technology [4]SmartMore

Project page: https://julianjuaner.github.io/projects/jenga

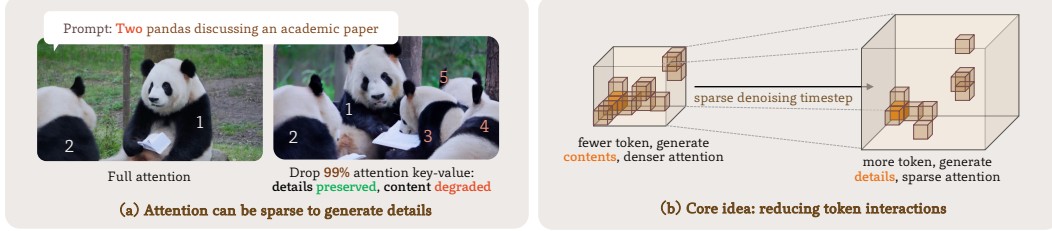

Figure 1: **Jenga generates high-quality videos with an efficient DiT inference pipeline.** *(a):* Extremely sparse attention can preserve details in generated videos. *(b):* We minimize token interactions via dynamic sparse attention with a progressive resolution design. We present videos generated by Jenga (sub-sampled 48 frames) among different models, marked with the DiT latency and relative speedup rate. Please use **Adobe Acrobat Reader for a live video visualization**.

## Abstract

Despite the remarkable generation quality of video Diffusion Transformer (DiT) models, their practical deployment is severely hindered by extensive computational requirements. This inefficiency stems from two key challenges: the quadratic complexity of self-attention with respect to token length and the multi-step nature of diffusion models. To address these limitations, we present Jenga, a novel inference pipeline that combines dynamic attention carving with progressive resolution generation. Our approach leverages two key insights: (1) early denoising steps do not require high-resolution latents, and (2) later steps do not require dense attention. Jenga introduces a block-wise attention mechanism that dynamically selects relevant token interactions using 3D space-filling curves, alongside a progressive resolution strategy that gradually increases latent resolution during generation. Experimental results demonstrate that Jenga achieves substantial speedups across multiple state-of-the-art video diffusion models while maintaining comparable generation quality ($8.83\times$ speedup with 0.01% performance drop on VBench). As a plug-and-play solution, Jenga enables practical, high-quality video generation on

39th Conference on Neural Information Processing Systems (NeurIPS 2025).

modern hardware by reducing inference time from minutes to seconds—without requiring model retraining.

# 1 Introduction

The advancement of Latent Diffusion Models [1, 2, 3, 4, 5, 6, 7] has significantly propelled the development of image and video generation. Recently, Diffusion Transformers (DiT) [8, 9, 10, 11, 12, 13, 14, 15, 16] have emerged as the predominant architecture for foundation models due to their inherent scalability and superior generative capabilities. As high-resolution video generation techniques continue to advance and DiT-based models scale to unprecedented sizes, the computational efficiency of generating high-quality content has become critically important. For example, generating a mere 5-second 720P video using HunyuanVideo [12] on a single NVIDIA H800 GPU requires approximately 27 minutes, severely limiting its practical applications in real-world scenarios.

This challenge stems from two orthogonal factors: (1) Self-Attention versus massive token length $N$. The continuously increasing token length for high-resolution generation causes a computational bottleneck due to the $O(N^2)$ computational complexity of self-attention in Transformers. Even with efficient attention mechanisms [17], self-attention in HunyuanVideo [12] still consumes 77.8% of the total processing time. (2) The multi-step nature of Diffusion models. The denoising process requires forwarding through the DiT architecture $T$ times, introducing $T$-fold computational overhead compared to non-diffusion models [18, 19] of similar specifications.

To address these challenges, various approaches have been explored. One branch focuses on operator-based acceleration, particularly attention optimization, to eliminate computational bottlenecks. STA [20], CLEAR [21], and SVG [22] predefine head-aware attention sparsity patterns in temporal or spatial dimensions. However, these approaches inadequately account for dynamic variations in attention patterns across inputs and achieve only modest speedup ratios (1.5–2×), insufficient for practical deployment. Orthogonal approaches optimize the diffusion generation pipeline through distillation [23, 24, 25, 26], quantization [27, 28, 29], or feature reuse [30, 31, 32]. However, distillation incurs significant training costs while often degrading output quality. Similarly, feature reusing and quantization methods also face limitations in achieving adequate acceleration ratios necessary for practical applications.

Based on the two orthogonal factors identified, we propose *Jenga*, a progressive, fully sparse inference pipeline with a dynamic, generalizable Attention Carving kernel. Studies have shown that the diffusion denoising process progresses from low to high-frequency generation [33, 34], where earlier steps establish content structures while later steps refine details. The core idea of Jenga is: **Early denoising steps do not require high-resolution latents, while later steps do not require dense full attention**. Once video content is established, the inherent redundancy in video latents means that not every token must participate in attention computations; at high resolutions, attention is inherently sparse, and fine details can be generated without full attention. Accordingly, Jenga designs a device-friendly Attention Carving kernel that decomposes latents into contiguous latent blocks using space-filling curves, and employs block-wise attention to selectively compute key-value pairs, creating an efficient attention mechanism. As illustrated in Fig. 1 (a), video details can be preserved even when we only keep 1% key-value blocks using Attention Carving.

Generating content layouts does not require huge latent inputs, so we introduce a multi-stage Progressive Resolution (ProRes) strategy that generates video through phased resizing and denoising of latents, effectively reducing the token interactions. Under this strategy, we face the challenge of generating resolution-dependent variations in the field of view that affect content richness. For example, low-resolution generation focuses on zoomed-in details rather than global scenes. To counteract this, we introduce a text-attention amplifier that reduces local neighborhood focus, enhancing condition information utilization, and producing more content-informative results, which are similar to generating content directly using high-resolution.

As illustrated in Fig. 1(b), Jenga is a combination of two complementary techniques: ProRes handles robust content generation with lower resolutions, while Attention Carving processes sparse attention, reducing token interactions. Like optimally arranged real-world Jenga blocks, these techniques deliver efficient, high-quality video generation with high block sparsity. Empowered by Jenga, we achieve impressive results across multiple state-of-the-art DiT-based video diffusion models. For instance, we obtain 4.68–8.83× speedup on HunyuanT2V [12] while maintaining comparable performance

on VBench [35]. Similarly, we demonstrate significant acceleration on HunyuanVideo-I2V (4.43×), the distilled model AccVideo [25] (2.12×), and Wan2.1 1.3B [13] (4.79×). Further, when deployed on an 8×H800 GPU computing node, Jenga reduces the DiT inference time to 39 seconds for HunyuanVideo and 12 seconds for AccVideo.

Our contributions are threefold: (1) we propose a novel dynamic block-wise attention carving approach that enables high-efficiency sparse attention computation for video generation; (2) we introduce Progressive Resolution, which decouples the content generation and detail refinement stages, reducing token interactions and achieving further acceleration; and (3) as a plug-and-play inference pipeline, Jenga achieves unprecedented speedup across various modern video DiT architectures.

## 2 Related Works

**Efficient attention design in Transformers** represents a critical research direction focused on mitigating computational demands arising from the quadratic $O(N^2)$ complexity relative to the token sequence length $N$. In Language Models, efficient attention methods like MInference [36], HIP [37, 38], MoBA [39], and NSA [40, 41, 42, 43] adopt partial or hierarchical key-value selections for efficient long-context understanding. To process dense vision features, efficient attention designs are also adopted in ViT and Diffusion Models, including linear attention [44, 45] and cascade attentions [46]. All these approaches aim to reduce the number of tokens actively participating in attention computations, thereby achieving acceleration and decreasing memory requirements.

**Efficient video generation** has garnered substantial research interest concurrent with the rapid evolution of video Diffusion Transformers (DiTs) [8, 11, 12, 47, 13]. Early acceleration techniques focused on reducing sampling steps, primarily through step distillation methodologies [25, 26] or training-free approaches that leverage step-wise feature reuse, such as TeaCache [31] and RAS [32, 48]. Bottleneck Sampling [49] employs a variable resolution strategy across different sampling stages, thereby utilizing fewer tokens during intermediate computational phases. Complementary to step reduction strategies, various efficient attention mechanisms for DiTs have emerged, including CLEAR [21], STA [20], and SVG [22], which operate on the fundamental assumption of localized attention distribution patterns. While this localization assumption preserves consistent attention structures, it inherently constrains the model's capacity for long-range feature aggregation. Recent advancements in block-wise attention architectures, such as SpargeAttn [50, 27] and AdaSpa [51], implement selective processing based on block-level mean values, achieving approximately two-fold acceleration in video generation pipelines. Nevertheless, their optimization potential remains limited by rigid block partitioning structures and attention sparsity parameters that require further finetune.

## 3 Jenga: Token-Efficient Optimization for Video Diffusion Transformers

Latent Diffusion Models (LDMs) [1] learns to reverse a noise corruption process, transforming random noise into clean latent-space samples. At time step $t \in \{0, \ldots, T\}$, the model predicts latent state $x_t$ conditioned on $x_{t+1}$: $p_\theta(x_t|x_{t+1}) = \mathcal{N}(x_t; \mu_\theta(x_{t+1}, t), \sigma_t^2 I)$, where $\theta$ represents the model parameters, $\mu_\theta$ denotes the predicted mean, and $\sigma_t$ is the predetermined standard deviation schedule. For Diffusion Transformers [8], during each timestep $t$, the model processes noisy visual latent tokens $x_t$ together with tokenized conditional embeddings $x_c$ (e.g., text prompt), predicting the noise component $\epsilon$ added at that timestep. A scheduler [52] then guides the progressive denoising process to compute the next denoised state $x_{t-1} = \mathbf{scheduler}(x_t, \epsilon, t)$, gradually yielding a fully denoised video latent $x_0$, which is then converted back to pixel space with a pre-trained VAE decoder.

The overview of our method is illustrated in Fig. 2. Jenga aims to minimize the computational complexity by reducing the number of tokens processed in each operation within video DiTs [8]. This is achieved through two primary optimizations: (1) enhancing the efficiency of the self-attention mechanism (Sec. 3.1) and (2) streamlining the inference pipeline (Sec. 3.2). In video DiT, we typically process $N_v = \mathrm{numel}(z_v) = \mathbf{thw}$ visual tokens, where $\mathbf{t}$, $\mathbf{h}$, and $\mathbf{w}$ represent the temporal length, height, and width of the video latent $z_v$ in the latent space, after the visual patch embedding layer, $z_v = \mathrm{patchemb}(x_v)$.

### 3.1 Block-Wise Attention Carving

As observed in [20, 50], the proportion of time spent on self-attention operations within transformer forward passes becomes increasingly dominant as the number of tokens grows. The 3D full-attention

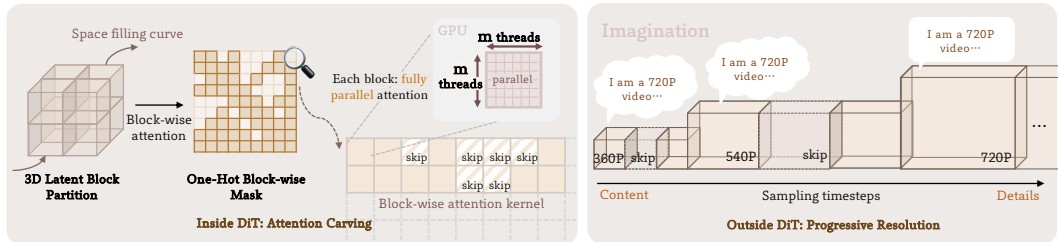

Figure 2: **Overview of Jenga.** *The left part* illustrates the attention carving. A 3D video latent is partitioned into local blocks before being passed to the Transformer layers. A block-wise attention is processed to get a head-aware sparse block-selection masks. In each selected block, dense parallel attention is performed. *The right part* illustrates the Progressive Resolution strategy. The number of tokens and timesteps is compressed to ensure an efficient generation.

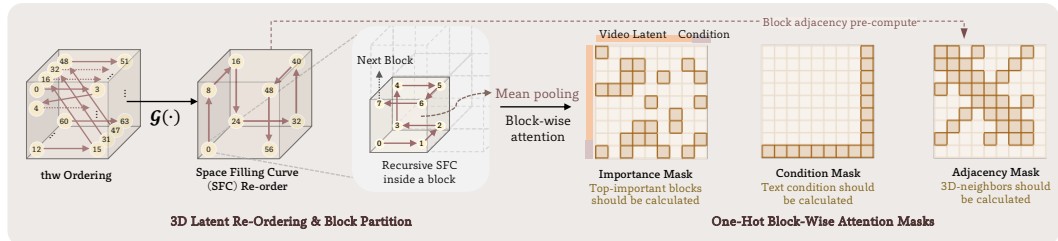

Figure 3: **Attention Carving (AttenCarve).** Here we illustrate a toy example of a $4 \times 4 \times 4$ latent, where $m = 8$ latent items form a block. *Left:* The latent 3D re-ordering and block partition via space filling curves (SFC). *Right:* After the block-wise attention in Eq. (3), we can construct the Importance Mask, combined with the pre-computed Condition Mask and Adjacency Mask, a block-wise dense attention mask is passed to the customized kernel for device-efficient attention.

mechanism in video transformers can be represented in its most fundamental form as:

$$\text{Attention}(Q_i, K_i, V_i) = \textbf{softmax}\left(Q_i K_i^\intercal / \sqrt{d_k}\right) V_i \, , \qquad (1)$$

where $Q_i, K_i, V_i \in \mathbb{R}^{N \times d_k}$ represent the query, key, and value features for the attention head $i$, respectively. We define $d$ as the embedding dimension, and $h$ as the number of attention heads with $d_k = d/h$. $N = N_v + N_c$ denotes the total number of tokens, comprising $N_v$ vision tokens and $N_c$ condition tokens. In the context of video diffusion models, this attention operation incurs significant computational overhead due to its quadratic complexity $O(N^2)$ concerning the token count across spatial and temporal dimensions.

Due to the inherent redundancy in video latents, a direct approach to improve efficiency is to reduce the number of key-value pairs each query attends to. We adopt a block-wise coarse key-value selection method, as shown in Fig. 3. FlashAttention [53, 17] and other GPU-optimized approaches [50, 20] uniformly divide $Q$ and $KV$ into $M$ blocks with $m = N/M$ tokens each, corresponding to $m$ parallel threads in the attention computation, to compute exact attention results across all $M^2$ blocks through parallel processing. For simplicity, we assume $N_v$ and $N_c$ are padded lengths divisible by $m$. Our objective is therefore to reduce KV pairs at the block level. First, to obtain tokens with higher internal similarity within 3D blocks, we reorder the 1D vision tokens $z_{\text{thw}}$ (flattened along $\textbf{thw}$ dimensions) into a block-wise order $z_{\text{blk}}$ before subsequent partitioning. The reordering and its inverse process are represented by:

$$z_{\text{blk}} = \mathcal{G}(z_{\text{thw}}) \, , \, z_{\text{thw}} = \mathcal{G}^{-1}(z_{\text{blk}}), \qquad (2)$$

where $\mathcal{G}(\cdot)$ represents an index permutation function implemented via the Generalized Hilbert re-ordering [54, 55, 56], a toy example of which is illustrated in the left part of Fig. 3. Compared with vanilla linear $\textbf{hwt}$ ordering, this space-filling curve (SFC) ordering ensures that tokens in 1D proximity within $z_{\text{blk}}$ effectively preserve their 3D neighborhood relationships from the original space. Thus, this approach enables uniform partitioning directly in the flattened dimension when computing attention operations.

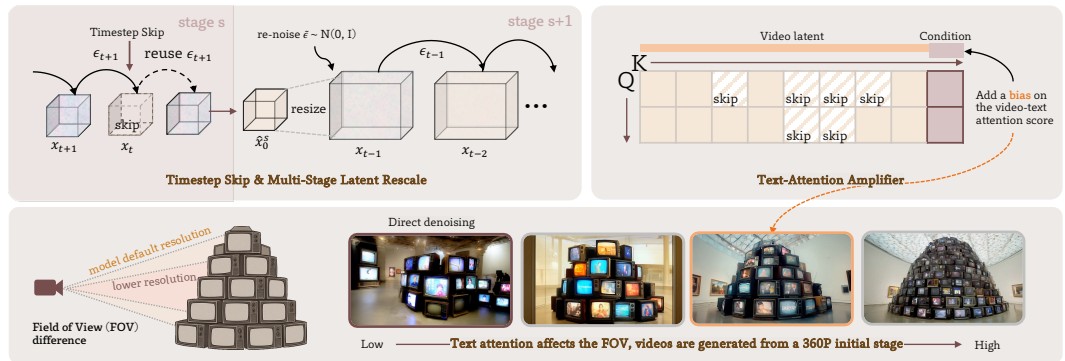

Figure 4: **Progressive Resolusion (ProRes).** *Left:* A brief illustration of stage switch and timestep skip. Before the rescale in stage $s$, we revert the latent to a clean state $\hat{x}_0^s$, then re-noise on the upsampled clean latent. *Right & Bottom:* We add a bias on the video-text attention score, to enable a scalable Field of View (FOV) in low-resolution content generation.

For KV-block selection, we build a one-hot block-wise 2D mask $\mathbf{B} \in \mathbb{R}^{M \times M}$ for each attention head to represent the selection result of the block-sparse attention. It is a union of three masks, as shown in the right part of Fig. 3: (1) Importance Mask $\mathbf{B}_{\text{top}}$. For importance-based block attention selection, inspired by MoBA [39] from large language models, we employ block-wise mean values to compute an attention probability map that roughly identifies which block pairs require attention computation. Specifically, for the reordered inputs, we express the relevance between blocks $\mathbf{R}$ using:

$$\mathbf{R} = \mathbf{softmax}(\hat{Q}\hat{K}^{\mathsf{T}}/\sqrt{d_k}), \tag{3}$$

where $\hat{(\cdot)}$ is a mean-pooling operator for each block of size $m$. Then for the $i$-th query block, we set a rate $k$, and keep the top $kM$ key-value blocks in $\mathbf{R}$. Meanwhile, for each query block, we set a constraint to fulfill the cutoff approximate accumulated probability. This means after all $kM$ blocks are selection, we still need to select blocks for some block-head combination with top probabilities, until the accumulated probability meets a cutoff softmax probability threshold $p$, defined by $\sum_{j \in \mathbf{B}_{\text{top}}[i]} \mathbf{R}[i][j] > p$. This constraint is set to avoid global context lost, especially for some attention heads to aggregate global information.

(2) Condition Mask. $\mathbf{B}_{\text{cond}} = \{i > N_v/m \lor j > N_v/m\}$, where $i, j$ are mask indices in query-key block dimensions. This means all condition-related attentions should be fully computed. (3) Adjacency Mask. $\mathbf{B}_{\text{adja}} = \{\mathbf{adja}(i,j)\}$, which represents whether $i$-th and $j$-th blocks are adjacent in the 3D **thw** space. The adjacency mask is beneficial in fixing border artifacts between spatially adjacent blocks. In Jenga, $\mathbf{B}_{\text{cond}}$ and $\mathbf{B}_{\text{adja}}$ are pre-computed, and only determined by the resolution and the partition function $\mathcal{G}$. The final selection array is defined as the union of three one-hot masks, $\mathbf{B} = \mathbf{B}_{\text{top}} \cup \mathbf{B}_{\text{cond}} \cup \mathbf{B}_{\text{adja}}$.

For the block-wise attention, we skip the computation of indices that $\mathbf{B}[i][j] = 0$, hence achieve an attention complexity $O(N'N)$, in which $N' = m \sum \mathbf{B}/M$ is the average number of selected tokens.

### 3.2 Progressive Resolution

Block-wise Attention Carving significantly reduces the latency of each DiT forward pass, but since diffusion sampling is an iterative process, compressing the number of tokens at the diffusion pipeline level is also crucial for accelerating generation. Leveraging the coarse-to-fine nature of diffusion denoising [33, 57], we decompose the generation inference process of $T$ timesteps into $S$ stages, starting from a low resolution $R_1 = \{\mathbf{t}, \mathbf{h}_1, \mathbf{w}_1, \mathbf{r}, \mathbf{d}\}$ and progressively increasing the resolution at each stage until reaching the final target resolution $R_S = \{\mathbf{t}, \mathbf{h}, \mathbf{w}, \mathbf{r}, \mathbf{d}\}$, where $\mathbf{r}$ represents the latent patch size and $\mathbf{d}$ is the channel dimension. The stage switch is illustrated in the left part of Fig. 4. At the end of each intermediate stage $s$ at timestep $t$, we predict the clean latent at $\hat{x}_0^s \in \mathbb{R}^{R_s}$ and resize it to a higher resolution $R_{s+1}$, then re-noised following an approach similar to [49]. The progressive resolution process between stages is defined as:

$$x_{t-1} = (1 - \sigma_t) \times \mathcal{U}(\hat{x}_0^s) + \sigma_t \tilde{\epsilon} \text{, where } \hat{x}_0^s = x_t - \sigma_t \epsilon_t \text{, } \tilde{\epsilon} \sim \mathcal{N}(0, I) \, . \tag{4}$$

Here $\mathcal{U}(\cdot)$ is a latent upsample function in 3D space, for which we employ area interpolation. $\epsilon_t$ is the prediction at timestep $t$, and $\sigma_t$ is the time-dependent standard deviation in the scheduler [52]. By reducing resolution, the earlier stages involve significantly fewer tokens in inference, while the denoising at higher resolutions ensures the generated videos maintain high-quality details.

**Text-Attention Amplifier.** Unlike bottleneck-style sampling [49], ProRes determines video content and structure during the low-resolution stage, without preserving the original resolution in the initial stage. While Video DiT generates coherent low-resolution videos, we observe that the Field of View (FOV) degrades with decreasing resolution, effectively transforming ProRes into a super-resolution process on videos with a constrained FOV. We illustrate this phenomenon in Fig. 4, which occurs because tokens at lower resolutions disproportionately attend to their spatial neighborhoods.

To maintain a stable FOV across resolutions, we introduce a text-attention amplifier with a resolution-dependent bias $\beta$ that "hypnotizes" the model in the first low-resolution stage by enhancing text-attention weights, thereby reducing the focus on spatial neighborhoods. This concept is illustrated in Figs. 2 and 4. Specifically, when processing a vision query block $q_v$ and a condition key block $k_c$ in attention, the biased vision-condition attention score is calculated as: $q_v k_c^\mathsf{T} + \beta$ where $\beta = -\rho \log(\mathrm{numel}(R_s)/\mathrm{numel}(R_S))$ is computed based on the token count ratio between the current stage resolution $R_s$ and the target resolution $R_S$, with $\rho$ serving as a balancing factor.

**Case-Agnostic Timestep Skip.** Timestep reduction is one of the most common optimization directions in efficient diffusion pipelines. Methods like TeaCache [31, 32] approximate outputs by caching input features to dynamically determine which steps can be skipped. However, in practical implementation, we observe that TeaCache's skip mechanism is effectively a static timestep scheduler, rather than a truly case-wise dynamic step skipping approach. Therefore, we employ a fixed timestep skip setting (23 steps, same as TeaCache-fast) that samples more densely at the beginning and end while sampling sparsely in the middle, eliminating the additional computation overhead of TeaCache.

## 4 Experiments

### 4.1 Implementation Details.

**Settings.** Our experiments are primarily conducted on the HunyuanVideo [12] architecture with a 50-step configuration. All generated HunyuanVideo videos maintain a resolution of $125 \times 720 \times 1280$, corresponding to a patchified video latent size of $\mathbf{t} \times \mathbf{h} \times \mathbf{w} = 32 \times 45 \times 80$, approximately 115K tokens. Unless specified, all experiments are performed on one NVIDIA H800 GPU.

For Attention Carving block partitioning, we employ Generalized Hilbert [54] as $\mathcal{G}(\cdot)$ with a block size of $m = 128$. We implement the Attention Carving kernel using Triton [58] and adopt a progressive top-K selection strategy when computing the importance mask: $k = 0.3$ at stage 1, and $k = 0.2$ for subsequent stages. The probability threshold is set to $p = 0.3$. When calculating the adjacency mask $\mathbf{B}_\mathrm{adja}$, it incorporates a 26-neighborhood in 3D latent space. For ProRes stages, we provide two basic configurations—Base and Turbo—corresponding to implementations using 1 (straight 720P) and 2 stages (starting with 540P, 50% steps each stage). We also introduce a 2-stage Jenga-Flash setting, which applies smaller $k$ values in both stages to further enhance efficiency. The balancing factor of the text-attention amplifier is set to $\rho = 0.5$. After timestep skipping, 23 of the original 50 timesteps are retained, while additional steps will be added after the stage-switch process. We adopt TeaCache-style [31] latent reuse, where features are reused before the image unpatchify layers. Comprehensive details are provided in Appendix B.1.

**Multi-GPU Adaptation.** Our method seamlessly integrates into multi-GPU parallel processing configurations. We have implemented adaptations based on xDiT [59, 60] within our approach using the HunyuanVideo [12] framework. This enables parallel processing of attention operations across the head dimension, while all operations except patchification are parallelized across multiple GPUs along the token dimension. Utilizing an 8-GPU parallel configuration, Jenga-Flash achieves a further $6.28\times$ speedup ($245s \to 35s$) with identical computational operations, which is also $5.8\times$ faster than the official 8-GPU implementation in HunyuanVideo [12]. Detailed latency results are shown in Tab. 2b. We provide specifications of this implementation in Appendix B.2.

**Distilled Model & Image-to-Video.** Jenga demonstrates considerable generalizability across diffusion model architectures. It not only achieves substantial acceleration ratios in Text-to-Video (T2V) models [12, 13], but our adaptive attention carving technique can also be effectively implemented in models refined through step-distillation [25, 26] with a $3.16\times$ speedup. Furthermore, when applied

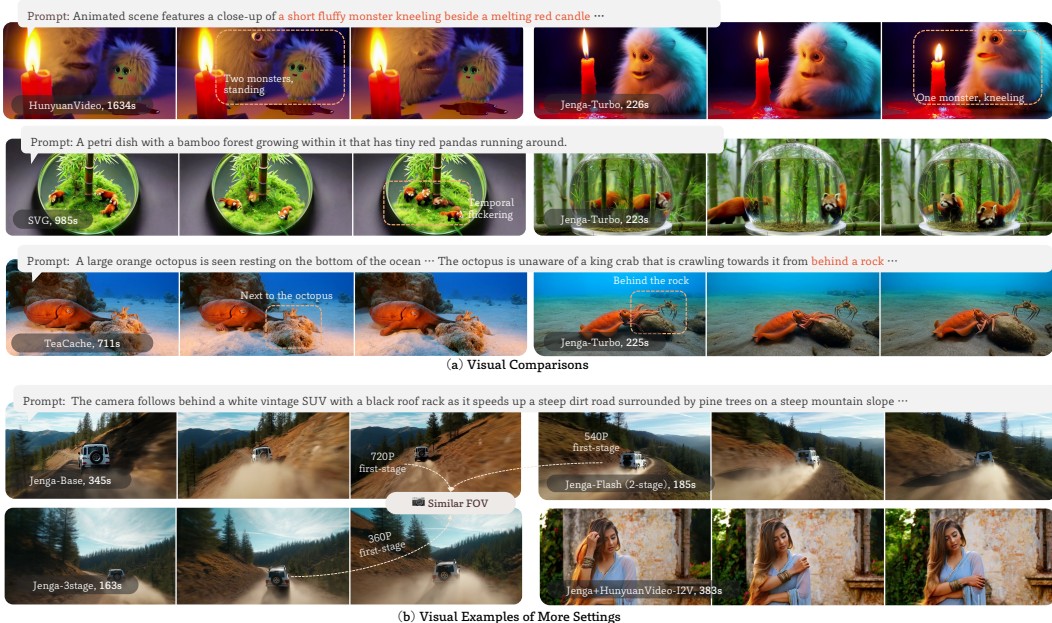

Figure 5: **Qualitative comparisons.** *(a):* Jenga maintains strong semantic performance while producing high-quality videos. *(b):* Examples across multiple Jenga settings, we also demonstrate how the text-amplifier stabilizes Field of View (FOV) across different initial resolutions.

to Image-to-Video (I2V) models [12], our approach achieves 4.43× speed improvement in I2V generation [12] tasks even without employing ProRes. Detailed results are shown in Tab. 2.

**Evaluation and Metrics.** For speed assessment, we report the diffusion time consumed—specifically the DiT forward pass time—as the VAE decoding component remains constant across all configurations. We also report FLOPs and step-wise FLOPs to provide an intuitive comparison of computational complexity. For qualitative evaluation, we employ the widely adopted CLIP-based metric CLIPScore [61] to measure text-video alignment, and utilize the comprehensive benchmark suites VBench [35] and VBench-I2V [62] with their original full-set prompts. We evaluate each prompt with a fixed random seed to ensure both evaluation consistency and statistical reliability. Additionally, we conducted a user study to assess human preference rates between Jenga and various efficient generation baselines, including a direct comparison with vanilla inference.

### 4.2 Comparisons

**Attention Efficiency.** We benchmarked our Attention Carving (AttenCarve) approach against state-of-the-art training-free attention optimization methods, specifically MInference [36], CLEAR [21], and SVG [22], as shown in Tab. 1. From a theoretical perspective, CLEAR (3D local window) and SVG (spatial-temporal windows) can be viewed as specialized instances of our more general Jenga framework. To establish a robust block-selection baseline, we adapted MInference [36] for video generation by removing causal masks and modifying selection optimizations. Jenga's dynamic block selection mechanism more effectively identifies crucial key-value pairs in video content while preserving important local information aggregation. Consequently, AttenCarve achieves superior acceleration ratios (2.17×) with reduced computational requirements while maintaining higher generation quality, particularly in terms of semantic adherence, compared to existing approaches.

**Sampling Efficiency.** We further compared our Progressive Resolution (ProRes) approach with TeaCache [31] in Tab. 1. We observe that ProRes and timestep skipping represent orthogonal solutions that address different aspects of efficient sampling. By incorporating ProRes, we achieve a significant reduction in step-wise FLOPs while maintaining high-quality video outputs. Qualitative evaluations confirm that our Progressive Resolution strategy effectively preserves generated video quality while substantially improving computational efficiency (3.28× speedup).

Table 1: **Evaluation results on HunyuanVideo [12].** We report evaluations of the baseline (row 1), attention optimization methods (row 2-4), pipeline optimization methods (row 5-8), and the combined results of Jenga (row 9-11). Here VBench-Q and VBench-S stand for Quality and Semantic metrics in VBench [35]. **Best** and the second best scores are highlighted.

| Methods | | Computation Loads | | Quality Evaluation [35, 61] | | | | Latency & Speed | |
| --- | --- | --- | --- | --- | --- | --- | --- | --- | --- |
| | NFE | PFLOPs↓ | PFLOPs / step↓ | VBench↑ | VBench-Q↑ | VBench-S↑ | CLIP-score↑ | DiT time↓ | Speedup↑ |
| HunyuanVideo [12] | 50 | 534.44 | 10.68 | 82.74% | 85.21% | 72.84% | 30.67 | 1625s | 1.00× |
| CLEAR (r=32) [21] | 50 | 479.97 | 9.60 | 82.68% | 86.06% | 69.17% | 30.43 | 1848s | 0.89× |
| MInference [36]ₙₒₙ₋cₐᵤₛₐₗ | 50 | 187.79 | 3.76 | 83.36% | 85.41% | 75.16% | 30.73 | 815s | 1.99× |
| SVG [22] | 50 | 243.36 | 4.86 | 83.11% | 85.87% | 72.07% | 30.63 | 988s | 1.64× |
| **AttenCarve** | 50 | 163.04 | 3.26 | **83.42%** | 85.31% | 75.85% | 30.60 | 748s | 2.17× |
| TeaCache-slow [31] | 31 | 331.35 | 10.68 | 82.53% | 85.64% | 70.09% | 30.42 | 967s | 1.68× |
| TeaCache-fast [31] | 23 | 245.84 | 10.68 | 82.39% | 85.51% | 69.91% | 30.39 | 703s | 2.31× |
| **ProRes** | 50 | 353.21 | 7.06 | 82.85% | **86.20%** | 69.43% | 30.03 | 1075s | 1.51× |
| **ProRes-timeskip** | 24 | 162.29 | 6.76 | 82.57% | 85.78% | 69.73% | 30.13 | 495s | 3.28× |
| *AttenCarve + ProRes* | 50 | - | - | 84.65% | 76.98% | 83.12% | 30.25 | 485s | 3.35× |
| *Jenga-Base* | 23 | 75.49 | 3.28 | 83.34% | 85.19% | 75.92% | 30.59 | 347s | 4.68× |
| *Jenga-Turbo* | 24 | 47.77 | 1.99 | 83.07% | 84.47% | 77.48% | **30.78** | 225s | 7.22× |
| *Jenga-Flash* | 24 | **32.97** | **1.37** | 82.73% | 84.01% | **77.58%** | 30.77 | **184s** | **8.83×** |

Table 2: **Model adaptation & parallel computing.** All latencies are DiT forward time. We evaluate VBench [35] on T2V models and VBench-I2V [62] for I2V models.

(a) *Jenga on HunyuanVideo-I2V [12]* (row 1-4) and *Wan2.1 [13]* (row 5-7), while we report a timestep skip result as the efficiency baseline.

| Methods | NFE | VBench | latency | speedup |
| --- | --- | --- | --- | --- |
| HunyuanI2V [12] | 50 | 87.49% | 1499s | 1.00× |
| + TimeSkip | 23 | 87.67% | 720s | 2.08× |
| + *Jenga* | 23 | 87.75% | 338s | 4.43× |
| Wan2.1-1.3B [13] | 50 | 83.28% | 115s | 1.00× |
| + TeaCache-fast [31] | 15 | 82.63% | 34s | 3.48× |
| + *Jenga* | 15 | 82.52% | 17s | 6.52× |

(b) *Jenga on distilled model [25]* (row 1-4) and *multi-GPU inference* (row 5-7). For multi-GPU, benchmark results are the same as Jenga-Flash in Tab. 1.

| Methods | # GPU | VBench | CLIP | latency | speedup |
| --- | --- | --- | --- | --- | --- |
| AccVideo [25] | 1 | 83.82% | 31.23 | 161s | 1.00× |
| + *Jenga* | 1 | 83.29% | 31.12 | 51s | 3.16× |
| + *Jenga-8GPU* | 8 | 83.29% | 31.12 | 7s | 23.00× |

| # GPU ₛₚₑₑ𝒹₋ᵤₚ ᵣₐₜₑ | 1 ₈.₈× | 2 ₇.₉× | 4 ₇.₀× | 8 ₅.₈× | VBench |
| --- | --- | --- | --- | --- | --- |
| HunyuanVideo [12] | 1625s | 844s | 440s | 225s | 82.74% |
| + *Jenga-Flash* | 184s | 107s | 63s | 39s | 82.73% |

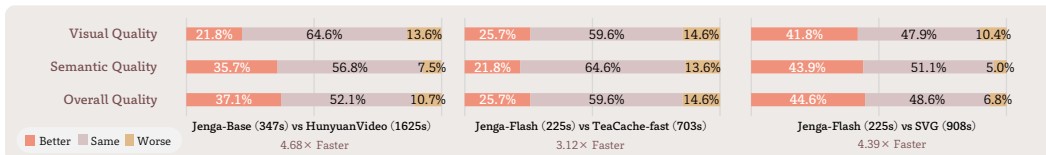

Figure 6: **User study.** We report pair-wise preference rates for visual, semantic, and overall quality.

**Qualitative Evaluation.** By orthogonally combining AttenCarve and ProRes with different stage configurations, we developed three variants of Jenga: Jenga-Base (1-stage), Jenga-Turbo (2-stage), and Jenga-Flash (2-stage, higher sparsity). These variants effectively balance generation quality and speed, achieving 4.68–8.82× acceleration while maintaining high-quality outputs. Notably, both Jenga-Base and Jenga-Turbo surpass the baseline on VBench [35] metrics, with particularly significant improvements in the semantic score (72.84% → 77.48%). This demonstrates that our approach not only accelerates inference but can also enhance the semantic coherence of generated videos. It is worth highlighting that when combining AttenCarve with timestep skipping alone (Jenga-Base), our quality metrics were not negatively affected. Jenga's focus on key information selection improves semantic performance. We provide visualization cases in Fig. 5. Meanwhile, our static case-agnostic timestep skip schedule performs similar behaviour to the TeaCache in both HunyuanVideo and HunyuanVideo-I2V [12]. Results are reported in Tabs. 1 and 2a.

**User Study.** We conducted a user study employing the standard win-rate methodology to evaluate our approach. Questionnaires were constructed, each containing 12 randomly selected videos generated using Sora prompts [63]. The videos were presented in randomized order, and participants were asked to evaluate them along three dimensions: visual, semantic, and overall quality. We collected a

Table 3: **Ablation Studies.** All latencies are DiT forward time.

(a) *Cutoff probability and multi-stage selection rates.* We report VBench / latency results. The second column and the head row represent drop rates in the first and second stages, respectively.

| $p$ | select rates $k$ | 0.3 | 0.2 | 0.1 |
|---|---|---|---|---|
| | 0.4 | 82.70% / 283s | 82.59% / 242s | 82.35% / 216s |
| 0.5 | 0.3 | 82.98% / 266s | 82.75% / 232s | 82.43% / 204s |
| | 0.2 | 83.07% / 253s | 82.89% / 222s | 82.60% / 198s |
| | 0.4 | 82.90% / 277s | 82.61% / 237s | 82.61% / 205s |
| 0.3 | 0.3 | 82.87% / 262s | **83.07%** / 225s | 82.96% / 195s |
| | 0.2 | 82.88% / 252s | 82.85% / 214 s | 82.73% / 184s |
| | 0.4 | 82.60% / 260s | 82.47% / 237s | 82.42% / 205s |
| 0.0 | 0.3 | 82.87% / 261s | 82.85% / 227s | 82.84% / 196s |
| | 0.2 | 83.01% / 248s | 82.85% / 212s | 82.67% / **183s** |

(b) *Block partition & block selection masks* (left and right are two separate tables, in row 1-3), *stage numbers* (row 4-7), and *text amplifier bias* (row 8-10). VB-Q/S represents VBench Quality and Semantic.

| *partition* | VBench | latency | *selection* | VBench | latency |
|---|---|---|---|---|---|
| hwt linear | 82.82% | 229s | w/o $\mathbf{B}_{\text{cond}}$ | 81.82% | 221s |
| SFC | 83.07% | 225s | w/o $\mathbf{B}_{\text{adja}}$ | 82.42% | 222s |

| *stage number* | VBench | VB-Q | VB-S | latency | speed |
|---|---|---|---|---|---|
| 1 (720P) | 83.34% | 85.19% | 75.92% | 347s | 4.68× |
| 2 (540-720P) | 83.07% | 84.47% | 77.48% | 225s | 7.22× |
| 3 (360-540-720P) | 80.53% | 81.66% | 76.00% | 157s | 10.35× |

| *bias factor* $\rho$ | -0.5 | 0.0 | 0.5 | 1.0 | 1.5 |
|---|---|---|---|---|---|
| VBench | 82.06% | 82.40% | **83.07%** | 82.87% | 82.80% |
| CLIP-score | 30.32 | 30.60 | 30.78 | 30.94 | **31.05** |

(c) *Ablation study on mask selection strategy.* We analyze the contribution of different mask components to overall performance. The "No Mask" baseline uses ProRes and timestep skip with full attention.

| mask type | $\mathbf{B}_{\text{top}}$ | $\mathbf{B}_{\text{cond}}$ | $\mathbf{B}_{\text{adja}}$ | VBench | latency | speed |
|---|---|---|---|---|---|---|
| No Mask (FA) | Full | Full | Full | 82.57% | 495s | 1.00× |
| TopK only | Part | | | 81.35% | 220s | 2.25× |
| TopK, SFC | Part | | | 81.41% | 220s | 2.25× |
| + Prob. Constraint | ✓ | | | 81.87% | 223s | 2.22× |
| w/o Adjacency | ✓ | ✓ | | 82.42% | 222s | 2.23× |
| w/o Condition | ✓ | | ✓ | 81.82% | 221s | 2.24× |
| w/o Importance | | ✓ | ✓ | 77.41% | **140s** | 3.54× |
| All Mask | ✓ | ✓ | ✓ | **83.07%** | 225s | 2.20× |

(d) *Ablation studies on key hyperparameters. (Top):* Progressive resolution design with varying low-res step ratios. *(Bottom):* Cutoff probability threshold analysis.

| Low-Res Step | 10% | 30% | 50% | 70% | 90% |
|---|---|---|---|---|---|
| Traj-Timestep | 987 | 947 | 883 | 767 | 437 |
| VBench | 82.36% | 82.35% | 83.07% | 81.03% | 78.74% |
| Latency | 286s | 253s | 225s | 207s | 169s |

| Cutoff Probability | 0.0 | 0.3 | 0.5 | 0.7 | 0.9 |
|---|---|---|---|---|---|
| VBench | 82.85% | 83.07% | 82.75% | 82.18% | 82.59% |
| Latency | 227s | 225s | 232s | 271s | 330s |
| Effective Sparsity | 80.3% | 80.1% | 78.5% | 72.3% | 57.5% |

total of 70 completed feedback forms, with results presented in Fig. 6. The findings demonstrate that our method is perceptually indistinguishable from multiple efficient generation baselines [22, 12, 31] when subjected to human evaluation.

## 4.3 Ablation Study and Discussions

To rigorously validate the effectiveness of our proposed method, we conducted comprehensive ablation studies on both Attention Carving and Progressive Resolution, with results in Tab. 3.

**Attention Carving.** As shown in Tab. 3a, we ablated selection rates $k$ and truncation probability $p$. Our results demonstrate robust performance even with a smaller $k$ (82.73% for 0.1-0.2 selection rate). The findings reveal a gradual decline in both latency and generation quality as selection rates increase in the second stage, while $k$ in the first stage has minimal impact on latency. The probability constraint enhances global information gathering, as illustrated in Fig. 7, but a large cutoff value ($p = 0.5$) disrupts the selection balance among attention heads, leading to slight performance degradation. Tab. 3b ablates latent-reorder and block selection strategies. Our experiments revealed that conventional linear partitioning can introduce shift artifacts in videos. Furthermore, this scanning approach disregards locality and consequently requires more blocks than space-filling curve (SFC) partitioning, resulting in marginally increased latency. Fig. 7 and Tab. 3b also validate the effectiveness of incorporating the adjacency mask $\mathbf{B}_{\text{adja}}$ and condition mask $\mathbf{B}_{\text{cond}}$, demonstrating their necessity.

For cutoff probability analysis in Tab. 3d, lower values (0.0-0.3) achieve higher effective sparsity (80.3%-80.1%) by selecting the most important blocks, while higher values approach full attention behavior. The effective sparsity significantly exceeds theoretical selection rates, revealing the inherently local nature of video attention patterns.

**Mask Selection Strategy.** Tab. 3c provides a comprehensive analysis of different mask components' contributions to overall performance. The results demonstrate that while the importance mask $\mathbf{B}_{\text{top}}$ alone achieves significant speedup (2.25×), incorporating probability constraints and space-filling curve (SFC) ordering provides marginal improvements. The condition mask $\mathbf{B}_{\text{cond}}$ and adjacency mask $\mathbf{B}_{\text{adja}}$ prove essential for maintaining generation quality, with their removal causing noticeable performance degradation. Notably, removing the importance mask entirely leads to substantial quality

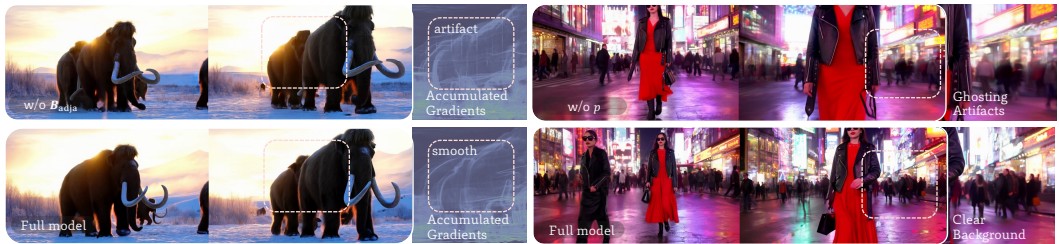

Figure 7: **Qualitative results for ablations.** *Left:* Missing Adjacency Mask $\mathbf{B}_{\text{adja}}$ causes grid effects on block borders. *Right:* Cutoff probability $p$ helps gain global contexts.

loss (77.41% vs. 83.07%), confirming its critical role in preserving video generation fidelity while enabling efficient sparse attention.

**Progressive Resolution.** The ablation studies presented in Tab. 3b demonstrate the effectiveness of our multi-stage approach. We found that a 2-stage configuration maintains strong generation quality, while increasing to $S = 3$ stages introduces some quality degradation due to latent alignment challenges. Nevertheless, the 3-stage variant still delivers satisfactory quality while achieving a $10.35\times$ speedup. Additionally, we evaluated the impact of various text-attention amplifier scales on generation quality. As shown in Tab. 3b, excessively high amplifier values introduce more global context and a shift in softmax distribution, resulting in some quality reduction. However, appropriately scaled amplifiers enhance content richness without compromising generation quality.

**Resolution Scheduling.** Tab. 3d reveals key insights about trajectory-guided resolution scheduling. The denoising trajectory serves as guidance for optimal resolution scheduling, with the 50% low-resolution step configuration achieving the best balance between quality (83.07%) and efficiency (225s). We observe dramatic quality degradation when low-resolution steps exceed 50%, while too few low-resolution steps also reduce quality, validating that low-resolution content generation requires higher attention density to establish proper structure.

**Limitation Analysis & Future Works.** While Jenga demonstrates compelling efficiency gains, some limitations remain. Foremost is maintaining latent alignment during resolution transitions–direct latent resizing offers computational advantages over pixel-domain operations (after VAE processes), but occasionally produces boundary artifacts. We found that these artifacts can be mitigated using detailed and comprehensive prompts. Our current implementation employs non-adaptive SFC block partitioning without leveraging semantic context for token importance, presenting a clear improvement opportunity. Future work could integrate learnable attention carving strategies during training rather than applying them post-hoc, potentially yielding optimal token selection while preserving Jenga's efficiency benefits. Detailed limitations are discussed in Appendix C.1.

## 5 Conclusion

In this paper, we introduce Jenga, a training-free inference pipeline that addresses computational bottlenecks in DiT-based video generation by dynamically managing token interactions. Our approach combines block-wise Attention Carving with Progressive Resolution, effectively decoupling content generation from detail refinement to significantly reduce computational complexity while preserving generation quality. Extensive experiments demonstrate substantial speedups up to $8.83\times$ across leading models, including text-to-video, image-to-video, and step distilled models. As a plug-and-play solution requiring no model retraining, Jenga represents a significant advancement toward making high-quality video generation more practical and accessible for real-world applications.

**Acknowledgements** This work was supported in part by the Research Grants Council under the Areas of Excellence scheme grant AoE/E-601/22-R. This work is partially supported by Hong Kong General Research Fund (14208023, 14206825), Hong Kong AoE/P-404/18, and the Centre for Perceptual and Interactive Intelligence (CPII) Ltd under InnoHK supported by the Innovation and Technology Commission.

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

# Training-Free Efficient Video Generation via Dynamic Token Carving

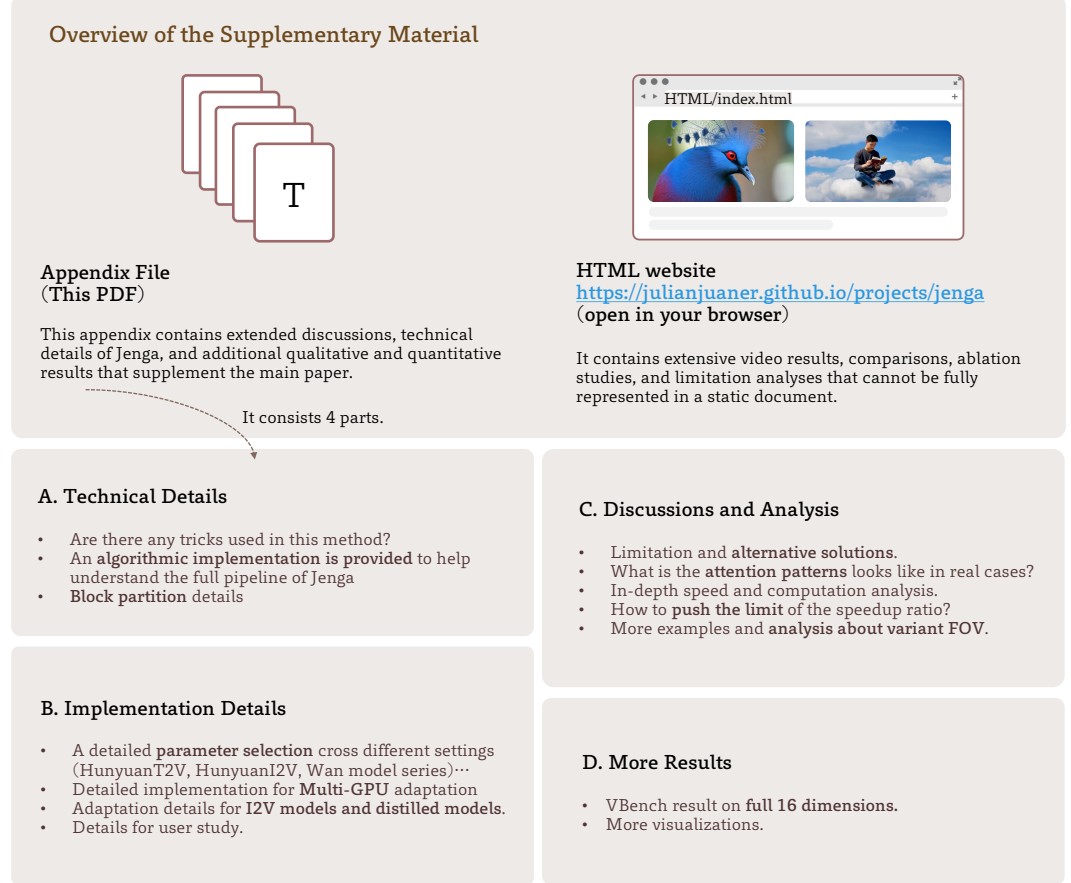

Figure 8: **Overview of the Supplementary.** We hope all readers enjoy this work in detail. We summarize common possible questions and important technical points here to arrange the supplementary. We strongly recommend that all readers open the link `https://julianjuaner.github.io/projects/jenga/` in your browser for video result visualizations.

## A   Algorithmic Implementation

For a more comprehensive understanding of the method component of Jenga, we provide pseudo-code algorithmic workflows in Algorithm 1 (Progressive Resolution), Algorithm 2 (Attention Carving pipeline), and Algorithm 3 (building block mask **B**).

### A.1   Details in Pipeline and ProRes

In the Progressive Resolution algorithm, we highlight three key technical details that were not fully elaborated in the main text.

- *Frequency re-ordering.* Prior to each attention layer, input latent patches undergo positional embedding operations such as RoPE [64], which typically establish frequency maps based on the standard **thw** ordering. Since we employ $\mathcal{G}$ to re-order the latents, we similarly apply $f_{\text{blk}} = \mathcal{G}(f)$ to re-order the frequency components $f$ across different dimensions, ensuring alignment with the latent ordering. As this operation is performed only once per stage, its computational overhead is negligible.

- *Ordering back before unpatchify.* Since the block selection in AttenCarve occurs after patchification, and both patchify and unpatchify operations need to be performed in the **thw** space,

---

**Algorithm 1** Progressive Resolution Framework for Jenga Video Generation

---

**Require:** Text prompt $c$, stage number $S$, resolutions $R_1, \ldots, R_S$, block size $m$, block selection rates $k_1, \ldots, k_S$, cutoff probability $p$, text-amplifier $\rho$, timestep lists $T_1, \ldots, T_S$
**Ensure:** Diffusion model $M_\theta$, flow-matching scheduler
1: Text tokens: $x_c = \text{LM}(c)$
2: **for** $s = 1$ to $S$ **do**
3:     Initial noise $\tilde{\epsilon} \sim \mathcal{N}(0, I) \in \mathbb{R}^{R_s}$, $x_T \leftarrow \tilde{\epsilon}$ **if** $s = 1$
4:     Compute block reordering $\mathcal{G}, \mathcal{G}^{-1}$ and adjacency masks $\mathbf{B}_{\text{adja}}$
5:     Remap positional frequencies: $f_{\text{blk}} \leftarrow \text{getFreq}(R_s, \mathcal{G})$
6:     **for** $t$ in $T_s$ **do**
7:         Reorder tokens: $z_t \leftarrow \mathcal{G}(\text{patchfiy}(x_t))$
8:         Apply sparse attention: $z_t \leftarrow M_\theta(z_t, x_c, k_s, \rho, f_{\text{blk}}, \mathbf{B}_{\text{adja}})$
9:         Restore order: $\epsilon_t \leftarrow \text{unpatchfiy}(\mathcal{G}^{-1}(z_t))$
10:         Denoise step: $x_{t-1} \leftarrow \text{scheduler}(x_t, \tilde{\epsilon}_t, t)$
11:     **end for**
12:     **if** $s > 1$ **then**
13:         Predict clean latent: $\hat{x}_0^s \leftarrow x_t - \sigma_t \epsilon_t$
14:         Resolution transition: $x_{t-1} \leftarrow (1 - \sigma_t) \times \mathcal{U}(\hat{x}_0^s) + \sigma_t \tilde{\epsilon}$
15:         Reset text amplifier: $\rho \leftarrow 0$ for $s > 1$
16:         Increase sampling shift: $\alpha \leftarrow \alpha + 2$
17:     **end if**
18: **end for**
19: **return** Final prediction $x_0$

---

---

**Algorithm 2** Block-Sparse Attention with Conditional Enhancement

---

**Require:** Query $Q$, Key $K$, Value $V$, top-$k$, block size $m$, text blocks $M_c$, probability threshold $p$, adjacency mask $\mathbf{B}_{\text{adja}}$
**Ensure:** Attention output
1: Get visual blocks $M_v \leftarrow \lfloor N/m \rfloor - M_c$
2: **if** $M_v > 0$ **then**
3:     Extract $Q_v$ from first vision blocks $\times M$ tokens
4:     $\mathbf{B} \leftarrow \text{BuildMask}(Q_v, K, k, p, M_c \cup \mathbf{B}_{\text{adja}})$
5:     $O_v \leftarrow \text{AttenCarve}(Q_{\text{normal}}, K, V, \mathbf{B})$
6: **end if**
7: **if** $M_c > 0$ **then**
8:     Extract $Q_c$ from remaining tokens
9:     $O_c \leftarrow \text{FullAttention}(Q_c, K, V)$: Text blocks see all.
10: **end if**
11: **return** $\text{concat}(O_v, O_c)$

---

---

**Algorithm 3** Build Block-wise Attention Mask

---

**Require:** Query $Q_v$, Key $K$, top-$k$, probability threshold $p$, visual blocks $M_v$, adjacency mask $\mathbf{B}_{\text{adja}}$
**Ensure:** Block selection mask $\mathbf{B}$
1: $\hat{Q}, \hat{K} \leftarrow \text{BlockPool}(Q_v), \text{BlockPool}(K)$, mean pooling per block.
2: Block attention scores: $\mathbf{S} \leftarrow \hat{Q}\hat{K}^\top / \sqrt{d_k}$
3: Convert to probabilities: $\mathbf{R} \leftarrow \text{softmax}(\mathbf{S})$
4: Sort probabilities: $\mathbf{R}_{\text{sorted}}, \mathbf{I} \leftarrow \text{sort}(\mathbf{R}, \text{desc} = \text{True})$
5: $\mathbf{C} \leftarrow \text{cumsum}(\mathbf{R}_{\text{sorted}})$
6: $N_k \leftarrow \max(\text{sum}(\mathbf{C} \leq p) + 1, k \cdot M_v)$
7: Initialize: $\mathbf{B}_{\text{top}} \leftarrow \text{zeros}(B, H, M_v, M_{\text{total}})$
8: Fill $\mathbf{B}_{\text{top}}$ using indices $\mathbf{I}[:, :, :, 0 : N_k]$

9: $\mathbf{B}_{\text{cond}} \leftarrow \{i > M_v \lor j > M_v\}$
10: $\mathbf{B} \leftarrow \mathbf{B}_{\text{top}} \cup \mathbf{B}_{\text{adja}} \cup \mathbf{B}_{\text{cond}}$
11: **return** $\mathbf{B}$

---

we must execute reordering after patchification. Subsequently, before unpatchification, we apply the inverse operation $\mathcal{G}^{-1}$ from Eq. (2), ensuring that all transformations are performed in the appropriate space.

- *Scheduler re-shift.* Following the re-noise process in Eq. (4), although theoretically we maintain the same noise strength, the clean state $\hat{x}_0^s$ still exhibits a discrepancy from the true distribution. To address this, we employ an approach similar to BottleNeck Sampling [49, 65], progressively increasing the timestep shift factor $\alpha$ of the rectified flow scheduler across stages.

## A.2   Details in AttenCarve

The implementation of AttenCarve builds upon the official codebase of block-wise MInference [36]. To enhance attention efficiency, we decoupled the vision and text query blocks as $Q = \text{concat}(Q_v, Q_c)$, and applied FlashAttention2 [17] directly to the condition blocks. For the cutoff probability constraint when constructing the importance mask $\mathbf{B}_{\text{top}}$, we formulate the optimization

**Algorithm 4** Block-Sparse Attention with Text Amplification Kernel

---

**Require:** Query $Q$, Key $K$, Value $V$, sequence lengths, qk scale, text amplifier $\rho$, text block start index, block mask $\mathbf{B}$, block dimensions
**Ensure:** Output features
1: start_m $\leftarrow$ program_id(0)      // Current query block
2: off_hz $\leftarrow$ program_id(1)      // Batch * head index
3: Load sequence length and check bounds
4: Initialize offsets for data loading
5: Load query block $q$ and scale by qk_scale
6: Initialize accumulators $m_i \leftarrow -\infty$, $l_i \leftarrow 0$, acc $\leftarrow 0$
7: **for** block_idx $= 0$ **to** NUM_BLOCKS $- 1$ **do**
8:     is_valid_block $\leftarrow \mathbf{B}[\text{off\_hz}, \text{start\_m}, \text{block\_idx}]$
9:     **if** is_valid_block **then**
10:         Load key-value block $k, v$ at offset block_idx $\times$ BLOCK_N
11:         Compute attention scores qk $\leftarrow q \cdot k^T$
12:         Apply sequence length mask to qk
13:         // Apply text amplification
14:         is_text_block $\leftarrow$ block_idx $\geq$ text_block_start
15:         qk $\leftarrow$ qk $+ \rho$ if is_text_block else qk
16:         Compute attention weights $p \leftarrow \exp(\text{qk} - \max(\text{qk}))$
17:         Update accumulators with standard attention updates
18:     **end if**
19: **end for**
20: Normalize: acc $\leftarrow$ acc$/l_i$
21: Write results to output

---

problem as minimizing the number of selected blocks:

$$\min_{\mathbf{B}_{\text{top}}[i]} |\mathbf{B}_{\text{top}}[i]| \quad \text{subject to} \quad \sum_{j \in \mathbf{B}_{\text{top}}[i]} \mathbf{R}[i][j] > p \tag{5}$$

To satisfy this constraint, our implementation employs a sort-then-greedily-select approach. For block index selection operations, we leverage vectorized indexing techniques to circumvent large-scale for loops, thereby substantially improving computational efficiency. In line 2 of Algorithm 3, we address an omission in the original Eq. (3) by explicitly incorporating the dimension $d_k$ in multi-head attention. Additionally, we implemented several engineering optimizations based on the MInference [36] block selection mechanism, including replacing the original `einsum` operations with CUBLAS-optimized `torch.bmm()` functions for enhanced latency performance.

### A.3   Index Re-Order and Block Partition

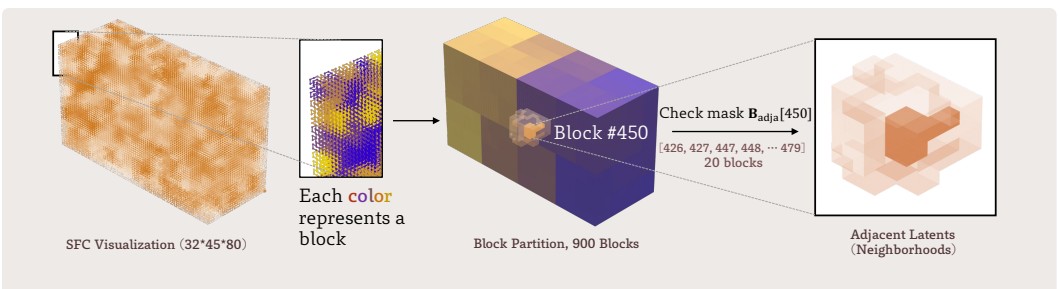

Figure 9: **A real block partition example.** We adopt a resolution-independent Space-Filling Curve (SFC) [54] to accommodate a wider range of resolutions compared to static 3D partitions. The right portion illustrates the local adjacent blocks using a look-up mask $\mathbf{B}_{\text{adja}}$.

To provide readers with a better understanding of the block partition characteristics in Jenga, beyond the toy example in Fig. 3, we demonstrate the Space-Filling Curve (SFC) implementation in a real 720P video latent space in Fig. 9. We employ Generalized Hilbert curves to overcome the limitation of standard Hilbert curves, which are only suitable for $(2^n, 2^n, 2^n)$ 3D spaces. It is important to

Table 4: **Detailed parameters.** We report the error bars for DiT latency measurements. The **bolded** steps indicate the additional steps required during stage transitions.

| Settings | NFE | AttenCarve | | S | ProRes | | | | Performance | |
|---|---|---|---|---|---|---|---|---|---|---|
| | | $k$ list | $p$ | | $R^s$ | step ratio | $\rho$ | $\alpha$ | latency | VBench |
| HunyuanVideo [12] | 50 | | | | $R^S =[32, 45, 80]$ | | | | $1625 \pm 15s$ | 82.74% |
| *Jenga-Base* | 23 | [0.3, 0.2] | 0.3 | 1 | $R^S \times [1.0, 1.0]$ | [0-24, 25-49] | 0.5 | [7] | $347 \pm 6s$ | 83.34% |
| *Jenga-Turbo* | 24 | [0.3, 0.2] | 0.3 | 2 | $R^S \times [0.75, 1.0]$ | [0-24, **25**-49] | 0.5 | [7, 9] | $225 \pm 5s$ | 83.07% |
| *Jenga-Flash* | 24 | [0.3, 0.2] | 0.3 | 2 | $R^S \times [0.75, 1.0]$ | [0-24, **25**-49] | 0.5 | [7, 9] | $184 \pm 3s$ | 82.73% |
| *Jenga-3Stage* | 24 | [0.3, 0.2, 0.2] | 0.3 | 3 | $R^S \times [0.5, 0.75, 1.0]$ | [0-14, 15-24, **25**-49] | 0.5 | [7, 9, 11] | $157 \pm 3s$ | 80.53% |
| HunyuanVideo-I2V [12] | 50 | | | | $R^S =[32, 45, 80]$ | | | | $1499 \pm 12s$ | 87.49% |
| *+ Jenga* | 23 | [0.3, 0.2] | 0.3 | 1 | $R^S \times [1.0, 1.0]$ | [0-24, 25-49] | 0.0 | [7] | $338 \pm 4s$ | 87.75% |
| AccVideo [25] | 5 | | | | $R^S =[32, 44, 78]$ | | | | $161 \pm 4s$ | 83.84% |
| *+ Jenga-Base* | 5 | [0.3, 0.2] | 0.3 | 1 | $R^S \times [1.0, 1.0]$ | [0-24, 25-49] | 0.5 | [7] | $76 \pm 2s$ | 83.39% |
| *+ Jenga-Turbo* | 5 | [0.3, 0.2] | 0.3 | 1 | $R^S \times [0.75, 1.0]$ | [0-24, 25-49] | 0.5 | [7] | $51 \pm 2s$ | 83.29% |
| Wan2.1-1.3B [13] | 50 | | | | $R^S =[20, 30, 52]$ | | | | $115 \pm 3s$ | 83.28% |
| *+ Jenga-Base* | 15 | [0.2, 0.1] | 0.9 | 1 | $R^S \times [1.0, 1.0]$ | [0-24, 25-49] | 0.0 | [7] | $24 \pm 2s$ | 82.68% |
| *+ Jenga-Turbo* | 15 | [0.2, 0.1] | 0.9 | 1 | $R^S \times [0.75, 1.0]$ | [0-24, 25-49] | 0.0 | [7] | $17 \pm 2s$ | 82.52% |

note that each block in Jenga is not a regular rectangular prism, but rather a local cluster of tokens that are naturally partitioned. This design provides Jenga with minimal constraints regarding video dimensions–without requiring padding along physical dimensions, it only necessitates that the total token count **thw** be divisible by the block count $m$. The continuity property of SFC in the original space also ensures a certain degree of semantic similarity among tokens within each block.

We further demonstrate how to utilize the Adjacency Mask $\mathbf{B}_{\text{adja}}$ to identify blocks that are spatially adjacent in 3D space based on their SFC representation. As illustrated, for block 450, by identifying the blocks to which neighboring tokens belong, we located 20 adjacent blocks that are subsequently incorporated into the attention computation for the current block.

## B Implementation Details

### B.1 Detailed Parameter Settings

In Tab. 4, we provide a comprehensive list of almost all key parameters used in this work. It is worth noting that although Jenga-Base employs a single-stage pipeline, we utilized different drop rates (0.7, 0.8) at different timesteps, effectively dividing our steps into two segments. We discovered that using a higher cutoff probability (i.e., $p = 0.9$) in Wan2.1 [13] significantly improved results without incurring additional computational time, suggesting the presence of a few attention heads that concentrate on global features. We briefly describe our ProRes adaptation specifically for HunyuanVideo [12] (i.e., Jenga-Base). We will implement ProRes adaptation for Wan2.1 [13] in the future.

### B.2 Multi-GPU Adaptation

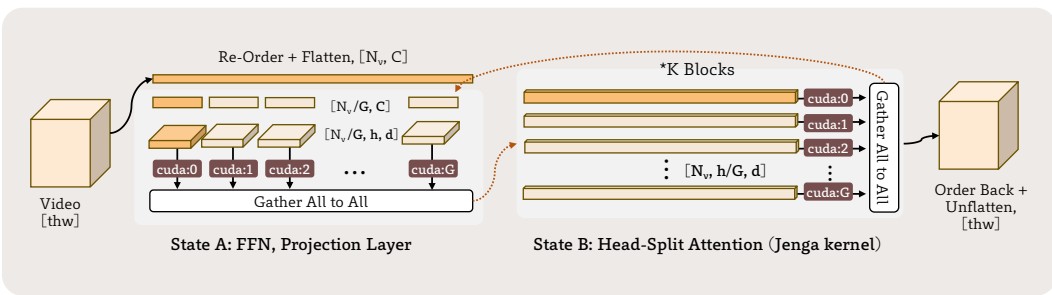

Figure 10: **Multi-GPU adaptation in Jenga.** We highlight the computation for each GPU in yellow.

For multi-GPU parallelism, we adapted our approach based on the xDiT [59] foundation used in HunyuanVideo. As illustrated in Fig. 10, we implemented parallelization across $G$ GPUs. The parallelism within Transformer blocks remains consistent with the original implementation (i.e., **state A**: parallelization along the token dimension before and after attention, and **state B**: parallelization along the head dimension within attention). We modified the corresponding LongContextAttention interface to make AttenCarve compatible with this parallel paradigm. Additionally, we discovered that when utilizing multi-GPU parallelism, the block selection process becomes the performance bottleneck. As explained in Appendix A.2, employing more efficient `torch.bmm` operations significantly accelerates multi-GPU execution (reducing processing time from 77s to 34s with 8 GPUs).

For parallelism outside transformer blocks, since we have naturally serialized tokens using SFC, we can directly partition them according to their SFC indices before feeding them into state A. This straightforward implementation also eliminates the previous requirement that latent sizes be divisible by $G$ along specific dimensions.

### B.3 Image-to-Video & Distilled Model

For Image-to-Video [12] adaptation, two specific details warrant clarification. Since this model performs specialized modulation operations on image conditions (latent at $\mathbf{t}[0]$), we provide an additional token-level mask $\mathcal{G}(\mathbf{m}), \mathbf{m} = \{1 \text{ if } \mathbf{t} = 0, \text{ else } 0\}$ when inputting tokens into the model. This enables decoupled modulation operations on the re-ordered latents. Additionally, the condition mask $\mathbf{B}_{\text{cond}}$ incorporates both text conditions and conditioning features from the first frame. Given that the first frame already contains the overall content of the video, we did not implement the text-attention amplifier.

For the distilled model AccVideo [25], which inherently requires fewer sampling steps, we employed a single-stage Jenga-Base setting as detailed in Tab. 4. Other configurations, including multi-GPU implementation, remain consistent with our HunyuanVideo setup.

### B.4 Compared Baselines

To establish a uniform evaluation standard, we standardized the test prompts, utilized the more widely adopted FlashAttention2 [17], and maintained consistent input video dimensions across experiments. Below are the specific configurations for comparison methods beyond the baseline:

- *CLEAR [21].* We implemented based on the original FlexAttention [66] with a 3D radius $r = 32$. When calculating FLOPs, since CLEAR does not account for GPU parallelism capabilities, we used the actual block sparsity (11.1% instead of the theoretical 56%) to compute effective FLOPs. Combined with the kernel optimization overhead of FlexAttention itself, the resulting generation speed could not even surpass the baseline.

- *MInference [36].* As explained in Sec. 4.2, we enhanced the block-wise attention mechanism from MInference. We removed the causal mask designed for LLMs and implemented a selection rate of $k = 0.3$. Notably, several approaches similar to MInference exist, such as block-sparse attention [67] and MoBA [39], which employ essentially identical methodologies.

- *SVG [22].* We utilized SVG's original implementation and resolution, incorporating its optimized RoPE and Normalization kernels with a sparsity setting of 0.2.

- *TeaCache [31].* We employed the official thresholds (0.1 for slow, 0.15 for fast configurations). For Wan2.1, we set the threshold to 0.2 and enabled the `use_ret_step` parameter, which provided further acceleration while preserving result quality.

### B.5 Details about User Study

Fig. 11 presents the Google Form questionnaire and anonymous website interface used to display video assets in our user study. We randomly sampled 12 prompts from a pool of 63 paired results and randomized the left-right ordering of videos within each comparison pair. To ensure data quality, we excluded invalid responses with completion times less than 5 minutes or greater than 1 hour. We also removed 3 submissions exhibiting highly homogeneous selection patterns (e.g., consistently choosing the "left video" or "same" for all comparisons). The results from the remaining 70 valid questionnaires are presented in Fig. 6.

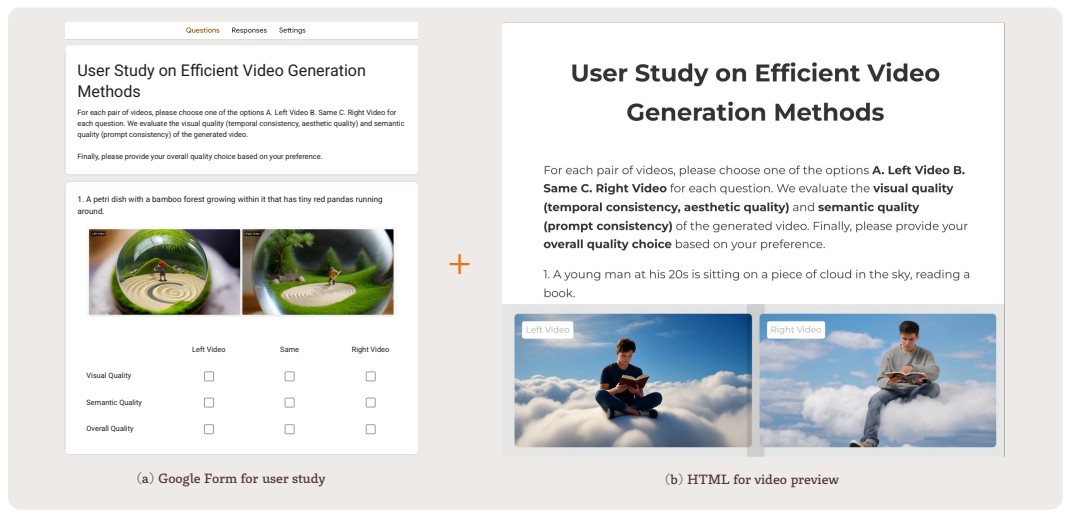

(a) Google Form for user study          (b) HTML for video preview

Figure 11: **User study.** *(a):* Questionnaire form example using Google Form. *(b):* Anonymous video preview website for live comparison.

## C    Discussions and Analysis

### C.1    Limitation Analysis & Alternative Solutions

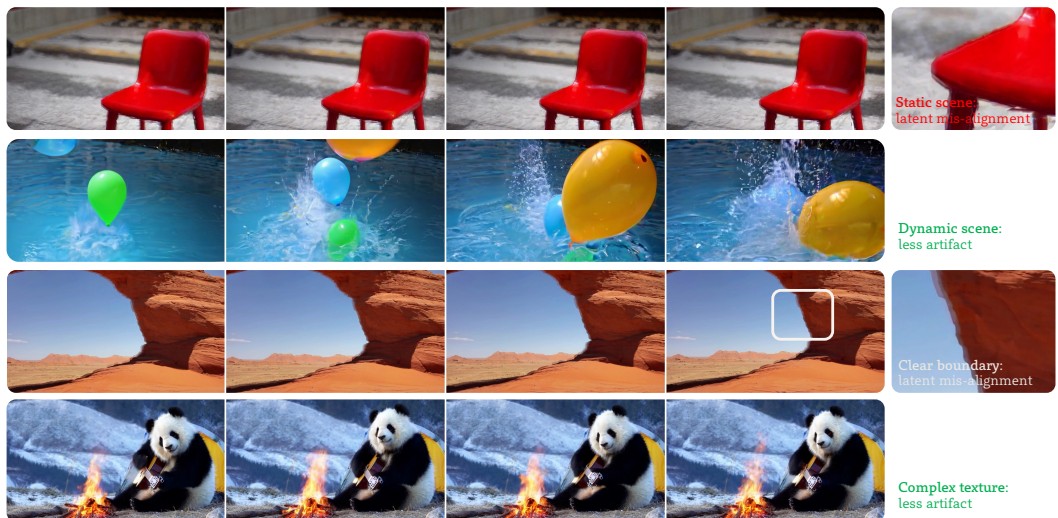

Figure 12: **Some failcases.** We present two potential failure cases that may occur when using more stages ($S > 3$), as well as scenarios where this setting is more suitable.

Table 5: **Results with different prompt formats.** Generation with enhanced prompts can eliminate quality degradation and boost multi-stage results (comparable video quality with $10.35 \times$ speedup).

| Prompt | HunyuanVideo [12] 1.00× | | | Jenga-Turbo (2-stage) 7.22× | | | Jenga-3Stage (3-stage) 10.35× | | |
|---|---|---|---|---|---|---|---|---|---|
| | VBench Total | VBench-Q | VBench-S | VBench Total | VBench-Q | VBench-S | VBench Total | VBench-Q | VBench-S |
| Standard | 82.74% | 85.21% | 72.84% | 83.07% | 84.47% | 77.48% | 80.53%-2.21% | 81.66%-3.55% | 76.00%+3.16% |
| Enhanced | 82.61% | 83.98% | 77.11% | 83.29% | 84.22% | 79.57% | 82.34%-0.27% | 83.65%-0.33% | 77.08%-0.03% |

As discussed in Sec. 4.3, Jenga faces certain challenges when implementing Progressive Resolution (ProRes). Several studies [68, 69] have examined the disparities between latent-space resizing and pixel-space resizing. Even with substantial re-noising ($\sigma_t > 0.9$), we cannot guarantee that edges in

the pixel space will be perfectly denoised in the final result. Since our work focuses on transformer acceleration, we opted against using the VAE decode-resize-encode approach, as tiled decode-encode operations during stage transitions would introduce additional latency of nearly 50 seconds. Fig. 12 illustrates some failure cases and usage scenarios of our current solution in 3-stage Jenga (results shown in Tab. 3b, 10.35× faster). We observed that generation quality occasionally deteriorates in static scenes or scenarios with clear boundaries (as well as in the Image-to-Video scenario). However, these issues tend to diminish when generating more complex textures or scenes with intricate motion patterns. We validated both the baseline and multi-stage results on VBench using enhanced prompts, as shown in Tab. 5. This enables users to obtain satisfactory video results with significant acceleration when using more complex prompts (such as Sora-style prompts, as demonstrated in Fig. 5 (b), the SUV case).

Beyond the training-based improvements discussed in Sec. 4.3, another promising direction for optimization is developing enhanced block partition methods. While the current SFC approach possesses many desirable properties, it remains fundamentally static. Extending context-based SFC approaches [70] into 3D video latent space could potentially yield better utilization of block selection.

## C.2 Block Selection: Attention Patterns

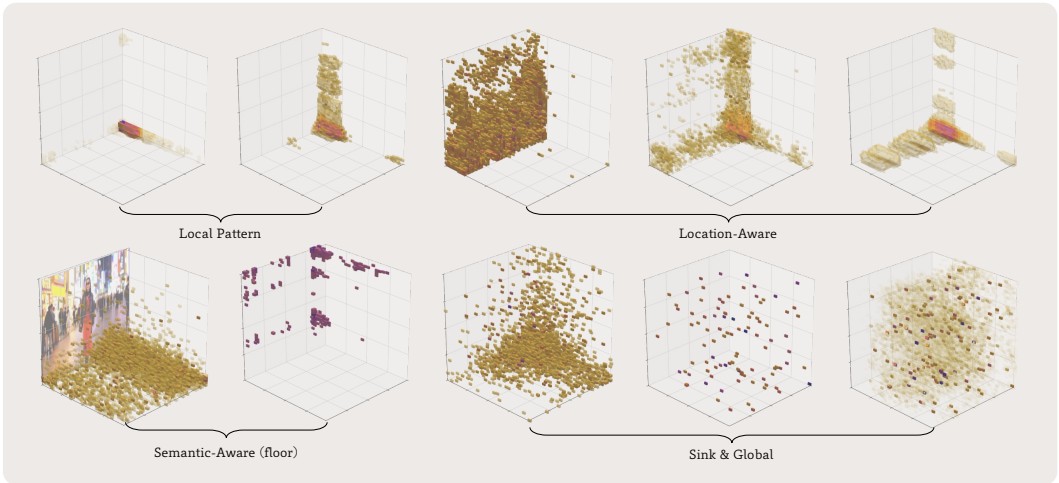

Figure 13: **Attention patterns.** Visualization of attention distributions across different layers and timesteps for the first block (at the corner position) containing 128 latent items.

We visualize the block-aware attention scores in Fig. 13. Our analysis reveals four key characteristics in the attention patterns: (1) In shallow layers, most patterns exhibit strong locality features, or (2) attention patterns highly correlate with position, forming stripe or planar distributions. In deeper layers of the model, (3) semantic-aware attention patterns emerge, where attention shifts according to the video's semantic content. (4) Simultaneously, we observe hybrid patterns combining the three aforementioned characteristics, as well as global patterns with attention sinks. Our cut-off probability threshold is specifically designed to capture information from these latter heads. These visualized patterns not only demonstrate the inherent sparsity characteristics of attention mechanisms but also highlight the necessity for dynamic block selection in our approach.

## C.3 Resolution-Aware Field of View

In addition to the influence of the text-attention amplifier on Field of View (FOV) demonstrated in Figs. 4 and 5, we present additional examples in Fig. 14 showing dynamic FOV changes achieved by adjusting the factor $\rho$. We observed that in certain scenarios, not utilizing the text-attention amplifier results in an overly localized focus, ultimately reducing the content coverage in the frame. By introducing the bias parameter $\beta$, we can exert a degree of control over different field-of-view ranges.

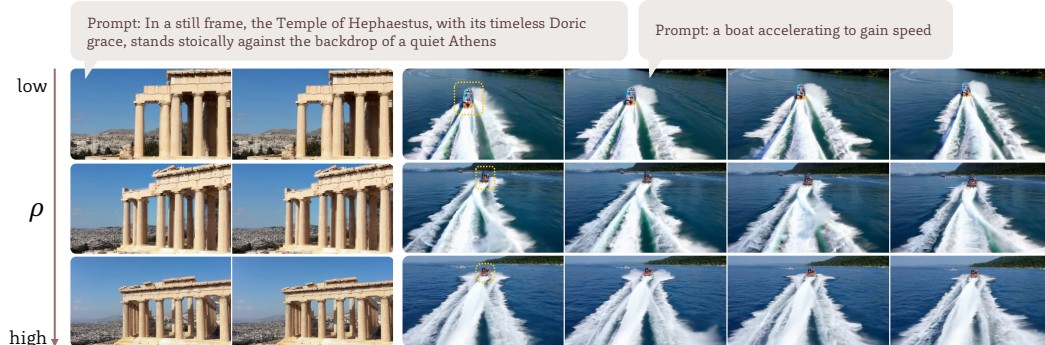

Figure 14: **Dynamic FOV.** We demonstrate the impact of the balancing factor $\rho$ on field of view in both static and dynamic scenes. Additional ablation examples are presented in the HTML supplement.

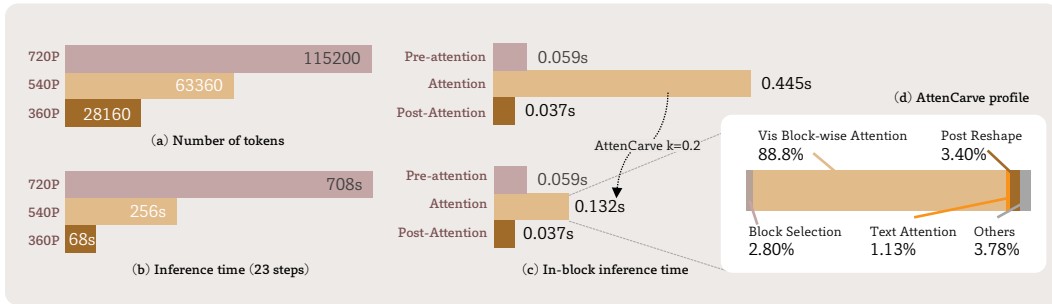

Figure 15: **Latency analysis.** *(a, b)* Visual token counts and generation times at different resolutions. *(c)* Acceleration of AttenCarve vs. FlashAttention2 [17]. *(d)* Time breakdown across AttenCarve components.

## C.4 Speed Analysis & Additional Overheads

In this section, we provide an in-depth analysis of our method's latency. First, as illustrated in Fig. 15 (a)-(b), we demonstrate the necessity of directly reducing token count by adjusting resolution. At 360P, only 1/4 of the input tokens, the generation speed achieves a $10\times$ improvement compared to 720P. In Fig. 15 (c), we specifically evaluate the acceleration achieved by AttenCarve compared to FlashAttention2 [17], which achieves a $3.7\times$ speedup in attention computation. Furthermore, Fig. 15 (d) provides a detailed time breakdown across different components of AttenCarve, showing that Block selection introduces only 2.8% computational overhead. Additionally, we analyzed the memory efficiency of our approach. Without any specialized optimizations, when generating 720P videos, Jenga introduces a minimal additional memory overhead of only 3.7% (71.84 → 74.49 GiB).

Despite the series of optimizations in Jenga, numerous avenues remain for potential performance improvements. These include incorporating quantization optimizations mentioned in SVG [22] and SpargeAttn [50], as well as kernel optimizations for RoPE [64] and normalization operations. From a hardware perspective, adapting FlashAttention3-based [71] attention kernels on the Hopper architecture shows significant speed enhancement potential. Additionally, parallelization and sparsification strategies for the VAE component have not been fully explored. These directions represent promising areas for future engineering optimizations and continued investigation in our work.

## C.5 Structural Fidelity Evaluation and Comparative Analysis

While our primary evaluation focuses on VBench for comprehensive video quality assessment, we recognize the value of complementary metrics that capture different aspects of video quality. Structural fidelity metrics like FVD (Fréchet Video Distance) provide additional validation of our method's effectiveness by evaluating the distance between training data distribution and inference data distribution. As a training-free method, we adopt two evaluation protocols: (1) the official VBench test results with different sampling seeds, and (2) the high-quality video dataset Inter4K [72] as real data distributions to evaluate FVD.

Table 6: **Structural fidelity evaluation with FVD metrics.** We compare Jenga with complementary acceleration methods across different models, demonstrating independent effectiveness.

| Method | VBench | Latency | Speed-Up | FVD-Inter4K ↓ | FVD-Hunyuan ↓ |
|---|---|---|---|---|---|
| HunyuanVideo | 82.74% | 1625s | 1.00× | 600 | 144 |
| AttenCarve Only | 83.42% | 748s | 2.17× | 448 | 164 |
| ProRes Only | 82.85% | 1075s | 1.51× | 620 | 191 |
| ProRes + Skip | 82.57% | 495s | 3.28× | 630 | 203 |
| AttenCarve + ProRes | 83.12% | 485s | 3.35× | 542 | 127 |
| *Jenga-Turbo* | **83.07%** | **225s** | **7.22×** | **583** | **141** |

| Method | VBench | Latency | Speed-Up | FVD-Inter4K ↓ | FVD-Wan2.1 ↓ |
|---|---|---|---|---|---|
| Wan2.1-1.3B | 83.28% | 115s | 1.00× | 788 | 182 |
| TeaCache-fast | 82.63% | 34s | 3.48× | 738 | 232 |
| AttenCarve Only | 82.96% | 71s | 1.62× | 674 | 169 |
| AttenCarve + ProRes | 83.56% | 52s | 2.21× | 579 | 189 |
| AttenCarve + Skip (*Jenga-Base*) | 82.80% | 24s | 4.79× | 702 | 184 |
| *Jenga-Turbo* | **82.52%** | **17s** | **6.52×** | **548** | **194** |

Table 7: **Comparison of different partitioning strategies.** We evaluate computational overhead for 720×1280×129f videos with block_size=128, demonstrating SFC's superior efficiency.

| Mask Type | STA Tiled Local (6,8,8) | Optimized Local (3,8,16) | 3D-SFC |
|---|---|---|---|
| Padding Tokens | 19,440 | 7,920 | **112** |
| Additional MatMul Computation | 35.32% | 13.78% | **0.19%** |

**Independent Acceleration Analysis.** To demonstrate that Jenga achieves substantial acceleration independently and can be combined with orthogonal methods, we conducted comprehensive comparisons with TeaCache [31], a feature reuse technique. Tab. 6 shows results on both HunyuanVideo and Wan2.1 models, revealing that our acceleration is fundamentally independent of feature caching techniques.

The results demonstrate that Jenga preserves video generation quality comparable to baselines while achieving superior acceleration. For HunyuanVideo, Jenga-Turbo achieves an FVD score of 141 versus 144 for the baseline, maintaining structural fidelity while delivering 7.22× speedup. Similarly, on Wan2.1, our method achieves competitive FVD scores across different evaluation protocols. The FVD results align with VBench trends, confirming that our acceleration techniques do not compromise generation quality.

Importantly, while TeaCache focuses on reusing computed features across timesteps, Jenga reduces computation through progressive resolution and selective attention while boosting generation quality. This orthogonality means the methods can potentially be combined for even greater acceleration benefits. Our results show that Jenga's acceleration is fundamentally independent of feature caching techniques, providing a complementary approach to efficient video generation that maintains high structural fidelity.

**Design Choice: Static SFC vs. Adaptive Partitioning.** Our choice of static Space-Filling Curve (SFC) construction is motivated by several key considerations that balance effectiveness and computational efficiency. First, SFC block partition exhibits inherent locality properties that align with the predominantly local characteristics of attention computation in high-resolution video data. The generalized Hilbert curves naturally possess local-neighborhood properties where neighbors on the 1D curve correspond to neighbors in 3D space, as validated in Tab. 3b where SFC improves both latency and performance while reducing line-wise drifting artifacts compared to linear partitioning.

Second, SFC demonstrates remarkable parameter insensitivity compared to hand-crafted strategies. While local window approaches (as in STA [20] and CLEAR [21]) restrict models to specific resolutions with dimensional padding severely impacting performance, SFC's 1D nature makes it insensitive to resolution and block size parameters. As shown in Tab. 7, our 3D-SFC requires only 112 padding tokens and 0.19% additional computation, compared to 19,440 tokens and 35.32% overhead for STA's tiled local windows, demonstrating superior efficiency.

Furthermore, adaptive partitioning faces fundamental challenges in text-to-video generation. Since generation starts from pure noise, extracting meaningful semantic information for adaptive token partitioning in early denoising timesteps is problematic. We experimented with dynamic approaches,

including changing SFC dimension scanning directions during interleaved forward attention passes to enhance block interactions (similar to Swin-window shift). This approach yielded no significant quality improvements while introducing substantial processing overhead ( 20s per video) due to memory discontinuity and defragmentation costs. Our static SFC approach avoids these overheads entirely through pre-computation while maintaining the locality benefits essential for efficient sparse attention. While this represents a limitation with room for future improvement, the current design effectively balances computational efficiency with generation quality.

# D   Additional Results

## D.1   Detailed Benchmarks

Table 8: **Detailed VBench [35] results.** We omit the percentage symbol % for better preview.

| Methods | Quality Metrics | | | | | | | Semantic Metrics | | | | | | | | |
| --- | --- | --- | --- | --- | --- | --- | --- | --- | --- | --- | --- | --- | --- | --- | --- | --- |
| | subject consistency | background consistency | temporal flickering | motion smoothness | aesthetic quality | imaging quality | dynamic degree | object class | multiple objects | human action | color | spatial relationship | scene | appearance style | temporal style | overall consistency |
| HunyuanVideo [12] | 96.59 | 98.06 | 99.63 | 99.54 | 61.11 | 72.23 | 60.83 | 82.03 | 68.75 | 94.00 | 93.75 | 78.86 | 38.60 | 20.51 | 23.22 | 26.54 |
| CLEAR [21] | 97.15 | 97.82 | 99.61 | 99.57 | 63.03 | 68.88 | 45.83 | 58.59 | 48.89 | 92.00 | 93.27 | 69.41 | 44.18 | 20.97 | 22.61 | 26.36 |
| MInference [36] | 94.90 | 97.66 | 99.41 | 99.47 | 61.62 | 69.78 | 65.27 | 75.00 | 83.08 | 88.00 | 93.75 | 77.18 | 42.28 | 20.80 | 23.08 | 27.17 |
| SVG [22] | 96.40 | 97.75 | 99.61 | 99.55 | 61.78 | 69.96 | 61.11 | 74.52 | 63.56 | 94.00 | 90.36 | 77.25 | 34.16 | 20.20 | 23.39 | 26.23 |
| **AttenCarve** | 95.94 | 97.85 | 99.30 | 99.18 | 62.47 | 69.09 | 70.83 | 86.71 | 73.02 | 93.00 | 90.67 | 75.45 | 47.17 | 19.50 | 23.43 | 26.36 |
| TeaCache-slow [31] | 96.70 | 97.89 | 99.30 | 99.49 | 61.54 | 69.18 | 59.72 | 67.24 | 63.41 | 88.00 | 85.19 | 72.09 | 36.11 | 20.05 | 23.11 | 25.80 |
| TeaCache-fast [31] | 96.68 | 97.79 | 99.32 | 99.50 | 61.42 | 68.59 | 56.94 | 64.08 | 64.71 | 90.00 | 85.99 | 71.22 | 36.26 | 20.12 | 23.12 | 25.77 |
| **ProRes** | 96.16 | 97.58 | 99.72 | 99.55 | 63.75 | 70.36 | 70.83 | 82.81 | 55.15 | 89.00 | 88.24 | 67.26 | 26.10 | 20.46 | 21.89 | 26.79 |
| **ProRes-timeskip** | 95.57 | 97.68 | 99.74 | 99.54 | 62.93 | 68.97 | 72.22 | 76.95 | 59.19 | 90.00 | 88.24 | 67.11 | 29.04 | 20.66 | 21.75 | 27.04 |
| *Jenga-Base* | 95.09 | 97.86 | 99.31 | 99.18 | 62.47 | 69.09 | 72.22 | 86.71 | 73.02 | 88.00 | 90.67 | 75.45 | 47.17 | 19.51 | 23.43 | 26.36 |
| *Jenga-Turbo* | 93.42 | 96.85 | 99.31 | 98.85 | 63.89 | 66.64 | 77.78 | 94.14 | 66.91 | 94.00 | 95.31 | 73.76 | 50.37 | 19.85 | 23.74 | 27.98 |
| *Jenga-Flash* | 92.75 | 97.19 | 99.27 | 98.57 | 62.29 | 66.71 | 85.71 | 73.61 | 63.60 | 90.00 | 99.26 | 71.97 | 56.25 | 20.27 | 24.43 | 28.05 |
| AccVideo [25] | 95.92 | 97.53 | 99.35 | 99.28 | 61.40 | 67.98 | 58.33 | 89.40 | 76.30 | 88.00 | 92.50 | 80.29 | 51.09 | 20.49 | 24.43 | 26.73 |
| *+Jenga* | 95.36 | 96.97 | 99.26 | 99.02 | 61.38 | 68.10 | 66.67 | 90.37 | 75.41 | 86.00 | 93.62 | 78.83 | 46.72 | 20.57 | 24.11 | 26.92 |
| Wan2.1-1.3B [13] | 96.46 | 98.40 | 99.52 | 98.72 | 64.08 | 67.36 | 59.72 | 75.00 | 47.64 | 82.00 | 81.87 | 71.49 | 23.11 | 19.82 | 23.68 | 23.59 |
| + TeaCache [31] | 96.40 | 98.25 | 99.38 | 98.70 | 62.03 | 65.59 | 58.33 | 76.39 | 47.48 | 78.00 | 82.47 | 69.16 | 24.13 | 19.83 | 23.14 | 22.99 |
| *+Jenga* | 95.40 | 97.92 | 99.44 | 98.55 | 61.13 | 65.37 | 61.11 | 74.76 | 53.89 | 78.00 | 88.42 | 70.08 | 26.53 | 20.25 | 23.34 | 23.49 |

| Methods | Quality Metrics | | | | | | | I2V Semantic Metrics | | | | Total |
| --- | --- | --- | --- | --- | --- | --- | --- | --- | --- | --- | --- | --- |
| | subject consistency | background consistency | motion smoothness | aesthetic quality | imaging quality | dynamic degree | **Quality Score** | camera motion | subject consistency | background consistency | **I2V Score** | **Total Score** |
| HunyuanVideo-I2V [12] | 95.67 | 96.39 | 99.21 | 61.55 | 70.37 | 21.14 | 78.30 | 51.38 | 98.90 | 99.38 | 96.67 | 87.49 |
| + timeskip | 95.75 | 96.86 | 99.22 | 61.93 | 70.84 | 21.54 | 78.64 | 51.51 | 98.92 | 99.42 | 96.71 | 87.67 |
| *+Jenga* | 93.99 | 95.75 | 99.00 | 60.84 | 70.43 | 40.65 | 79.31 | 49.80 | 98.43 | 99.14 | 96.18 | 87.74 |

Tab. 8 provides comprehensive evaluation results across all 16 dimensions of VBench [35]. As shown, Jenga achieves notable advantages in multiple semantic score dimensions while maintaining high performance in quality metrics.

Regarding detailed results in Tab. 8, there are two key points to clarify. First, we discovered that compared to the static local patterns used in CLEAR [21], our query/head-aware dynamic patterns significantly enhance the dynamic degree of generated results (45.83% → 70.83%). Overall, Jenga introduces larger motion amplitude at the quality level, while presenting some trade-offs in subject

consistency when the selection rate is small (Jenga-Flash). At the semantic level, Jenga demonstrates substantially better semantic adherence across multiple dimensions (color, object class, scene, and overall consistency).

### D.2 More Visual Results

We showcase additional results of Jenga in different settings, as illustrated in Fig. 16, and Fig. 17. We recommend viewing the video files in the provided HTML to better evaluate the effectiveness of our method.

## E Social Impacts

This paper introduces a novel framework for efficient video generation that is based on current pretrained Diffusion Transformers. Although this application has the potential to be misused by malicious actors for disinformation purposes, significant advancements have been achieved in detecting malicious generation. Consequently, we anticipate that our work will contribute to this domain. In forthcoming iterations of our method, we intend to introduce the NSFW (Not Safe for Work) test for detecting possible malicious generations. Through rigorous experimentation and analysis, our objective is to enhance comprehension of video generation techniques and alleviate their potential misuse.

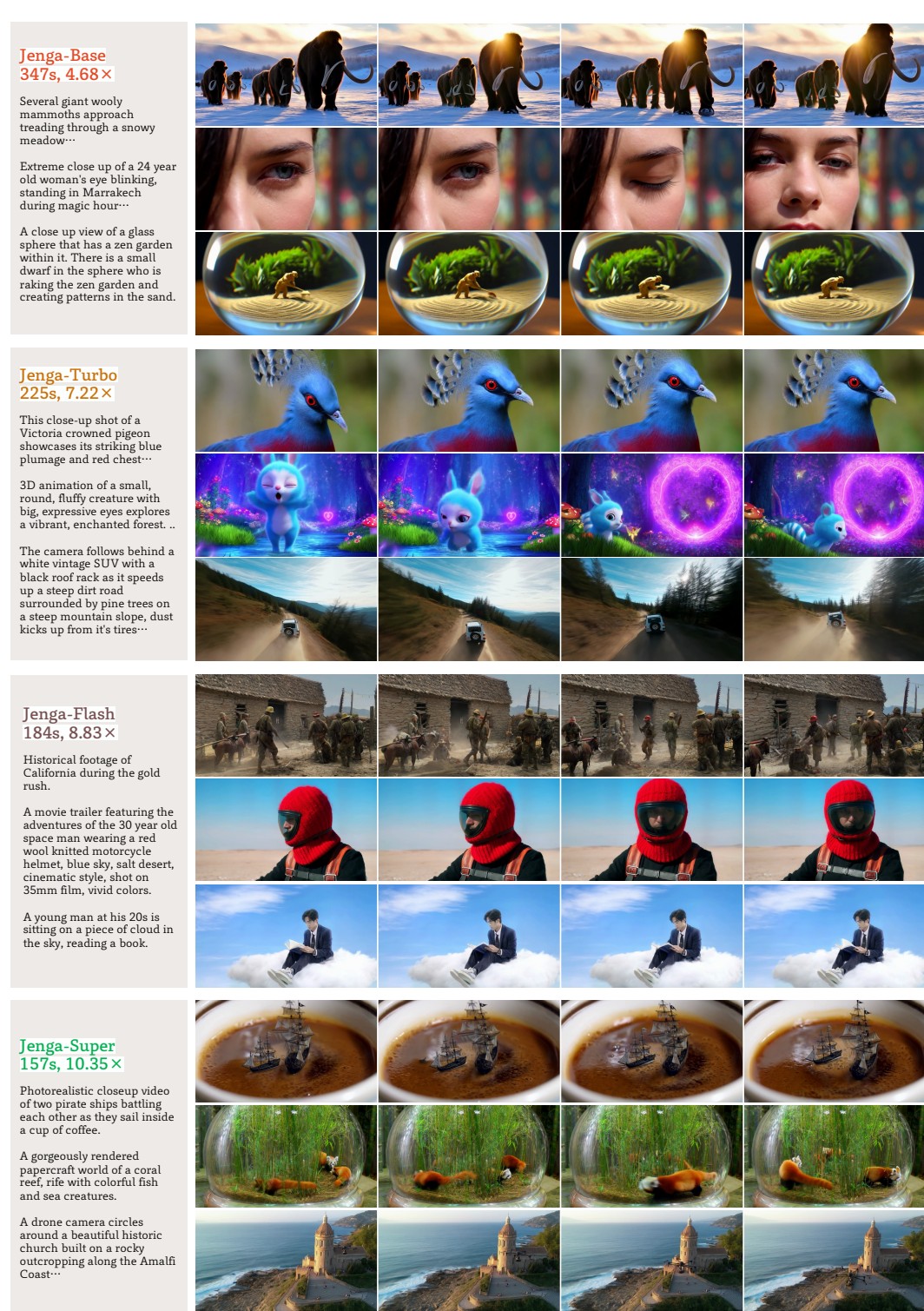

**Jenga-Base**
**347s, 4.68×**

Several giant wooly mammoths approach treading through a snowy meadow⋯

Extreme close up of a 24 year old woman's eye blinking, standing in Marrakech during magic hour⋯

A close up view of a glass sphere that has a zen garden within it. There is a small dwarf in the sphere who is raking the zen garden and creating patterns in the sand.

**Jenga-Turbo**
**225s, 7.22×**

This close-up shot of a Victoria crowned pigeon showcases its striking blue plumage and red chest⋯

3D animation of a small, round, fluffy creature with big, expressive eyes explores a vibrant, enchanted forest. ..

The camera follows behind a white vintage SUV with a black roof rack as it speeds up a steep dirt road surrounded by pine trees on a steep mountain slope, dust kicks up from it's tires⋯

**Jenga-Flash**
**184s, 8.83×**

Historical footage of California during the gold rush.

A movie trailer featuring the adventures of the 30 year old space man wearing a red wool knitted motorcycle helmet, blue sky, salt desert, cinematic style, shot on 35mm film, vivid colors.

A young man at his 20s is sitting on a piece of cloud in the sky, reading a book.

**Jenga-Super**
**157s, 10.35×**

Photorealistic closeup video of two pirate ships battling each other as they sail inside a cup of coffee.

A gorgeously rendered papercraft world of a coral reef, rife with colorful fish and sea creatures.

A drone camera circles around a beautiful historic church built on a rocky outcropping along the Amalfi Coast⋯

Figure 16: **Visualization results.** From top to bottom, each three videos is from the same setting.

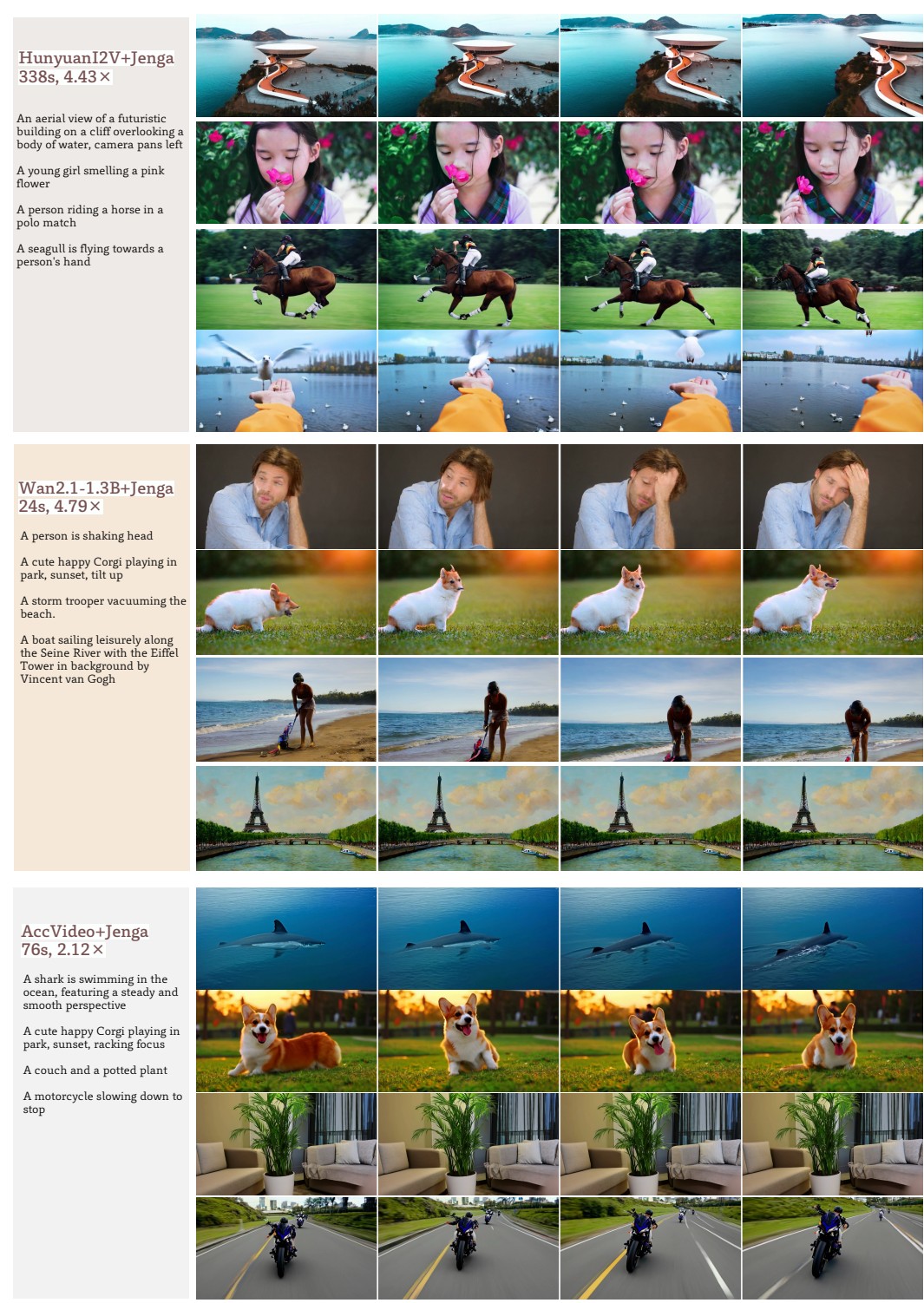

**HunyuanI2V+Jenga**
**338s, 4.43×**

An aerial view of a futuristic building on a cliff overlooking a body of water, camera pans left

A young girl smelling a pink flower

A person riding a horse in a polo match

A seagull is flying towards a person's hand

**Wan2.1-1.3B+Jenga**
**24s, 4.79×**

A person is shaking head

A cute happy Corgi playing in park, sunset, tilt up

A storm trooper vacuuming the beach.

A boat sailing leisurely along the Seine River with the Eiffel Tower in background by Vincent van Gogh

**AccVideo+Jenga**
**76s, 2.12×**

A shark is swimming in the ocean, featuring a steady and smooth perspective

A cute happy Corgi playing in park, sunset, racking focus

A couch and a potted plant

A motorcycle slowing down to stop

Figure 17: **Visualization results for model adaptations.** Prompts are from VBench [35].

