# OpenReview forum: "Training-Free Efficient Video Generation via Dynamic Token Carving"
_NeurIPS.cc/2025/Conference — NeurIPS 2025 poster_

### Official Review · Reviewer_cBBb · 2025-06-10

**Clarity:** 3
**Significance:** 3
**Originality:** 3
**Rating:** 4
**Confidence:** 3

**Summary:**

This paper presents a training-free and efficient video generation method called Jenga. It involves two aspects: dynamic attention carving and progressive resolution generation, which is inspired by the observed insight (1) early denoising steps do not require high-resolution latents, and (2) later steps do not require dense attention. The experiments are extensively conducted and the performance is superior, for example, x8.83 speedup with 0.01% performance drop on VBench.

**Questions:**

I am a little bit confused about the title and content. It seems the the title is highly related with attention carving. But Jenga involves progressive resolution generation. The two parts, attention carving and progressive resolution generation, are independent. Could the author explain it? In my view, it could be simply a combination of the two tricks.


It seems that the authors partially ablate different masking strategies. It could be better to increasingly add different masking strategies.

**Ethical Concerns:**

["NO or VERY MINOR ethics concerns only"]

**Limitations:**

The authors have highlighted the limitations.

**Quality:**

3

**Strengths And Weaknesses:**

Strengths:

Clear to follow

Extensive and logical experiment

Nice illustration

Weaknesses:

Latent alignment and artifacts during resolution transitions

Uses static space-filling curves (SFCs) for latent block partitioning

purely heuristic and post-hoc attention carving, without adaptability

---

> ### Author Rebuttal · Authors · 2025-07-28
>
> Dear reviewer cBBb,
>
> Thank you for your valuable feedback and constructive comments. We are grateful for your positive assessment of our work, particularly your recognition of its clarity and comprehensive experiments. We will address your comments below and incorporate the necessary revisions in the updated paper.
>
> ## Q1 Latent Alignment May Introduce Artifacts
>
> We acknowledge this limitation in our discussion (L302-305) and provide a **detailed discussion in Appendix C.1**. Indeed, latent alignment requires certain trade-offs as a training-free method, since pure pixel-level decode-encode operations consume approximately 40 seconds, which is unacceptable for a video generation acceleration framework. Nevertheless, we provide several mitigation strategies and discussions:
>
> **1. High noise levels minimize alignment impact from resize+renoise operations.** During early denoising timesteps when noise levels are substantial (883/1000 of the actual denoising trajectory, in **G2mf Q2**), the resize and renoise operations have minimal impact on latent alignment since the semantic content is not yet well-formed.
>
> **2. Noise shift compensation.** Discussed in **Appendix L339-342**, we introduce adaptive noise shift adjustments to maintain consistency across resolution transitions, helping preserve the underlying latent structure.
>
> **3. Enhanced semantic conditioning.** By incorporating richer semantic conditions during the alignment process (i.e., using enhanced prompts, with results in **Appendix Table 5** as validation), we can better preserve content coherence across different resolutions.
>
> **4. Faster VAE integration.** Future work can leverage faster VAE variants to reduce the computational overhead of pixel-level operations while maintaining alignment quality.
>
> ## Q2 SFC is Static
>
> Your observation is accurate. SFC construction is completed before denoising and only depends on resolution and dimension ordering (e.g., hwt or thw, representing different orderings). As acknowledged in our limitation section (L305-307), this approach has room for improvement. We explain the rationale behind Jenga's SFC choice and challenges with adaptive partitioning:
>
> **1. SFC block partition exhibits inherent locality properties.** Attention computation for high-resolution data predominantly exhibits local characteristics (e.g., STA, SeedVR). SFC naturally possesses local-neighborhood properties, where neighbors on the 1D curve correspond to neighbors in 3D space. We validated in Table 3(b) that SFC improves both latency and performance while reducing line-wise drifting artifacts.
>
> **2. SFC demonstrates remarkable parameter insensitivity.** Our Generalized Hilbert curves maintain locality while being insensitive to parameters. Hand-crafted block partition strategies (local windows in STA & CLEAR, Swin windows in SeedVR) restrict models to specific resolutions, with dimensional padding severely impacting performance. SFC's 1D nature makes it insensitive to resolution and block_size parameters. The table below shows additional padding and computational overhead for 720×1280×129f videos with block_size=128:
>
> |Mask Type|STA Tiled Local (6, 8, 8)|Optimized Local (3, 8, 16)|3D-SFC|
> |-|-|-|-|
> |Padding Tokens|19440|7920|112|
> |Additional MatMul Computation|35.32%|13.78%|0.19%|
>
> "Optimized Local" represents the optimal GPU-friendly block partitioning, still significantly larger than Jenga's 3D SFC.
>
> **3. Adaptive Partition: Cannot be defined at arbitrary timesteps.** Text-to-video generation starts from pure noise, making it challenging for adaptive token partitioning to extract meaningful semantic information. Determining adaptive partitions through latent semantic correspondence in early denoising timesteps is problematic.
>
> **4. Benefits of static approaches: Frequent reordering degrades generation speed.** We experimented with changing SFC dimension scanning directions during interleaved forward attention passes to enhance block interactions (similar to Swin-window shift). This approach yielded no significant quality improvements while introducing substantial processing overhead (~20s per video). Token reordering operations cause memory discontinuity and defragmentation overhead. **Adaptive block partition strategies would face similar challenges**, with additional overhead for constructing the adaptive strategy itself--an overhead that static SFC avoids entirely through pre-computation.
>
> ## Q3 Adaptability of the Attention Carving
>
> Following our discussion on block partitioning in Q2, we acknowledge that Attention Carving employs heuristic strategies. In practice, during Attention Carving, within each attention head, the block selection for each query block is adaptive and content-dependent. The importance mask $\mathbf{B}_\text{top}$ is dynamically computed based on the current latent features and query-key relationships, making the carving process responsive to the evolving content during generation. While the SFC partitioning strategy is static, the actual attention patterns and block selections are adaptive to the semantic content at each timestep and layer, ensuring that the most relevant spatial-temporal relationships are preserved during the carving process.
>
> As a pure training-free method, the current version avoids the need for offline tuning procedures. Jenga's significantly smaller search space makes it easily adaptable to new models without extensive hyperparameter tuning. It would be a future direction to design a metric to determine adaptive sparsity.
>
> ## Q4 Title: "Token Carving", It Seems Not Related With ProRes?
>
> ProRes fundamentally reduces token interactions as well. By reducing resolution (and thus the input token count), ProRes effectively reduces token interactions. Both ProRes and Attention Carving share the same core motivation: **reducing computational complexity through strategic token interaction reduction**. ProRes achieves this through progressive resolution scaling, while Attention Carving accomplishes this through selective attention masking. The "carving" metaphor aptly describes both processes--ProRes "carves" away lower-resolution details in the early stage, while Attention Carving "carves" away less important attention connections. Together, they form a unified framework for efficient video generation that operates across both input token dimensions and attention pattern dimensions.
>
> ## Q5 Two Methods are Independent (similar to iQ7G Q1.3)
>
> While ProRes and Attention Carving operate independently, the core innovation of Jenga lies in recognizing that **attention sparsity and progressive resolution are fundamentally complementary** techniques that address different aspects of the computational bottleneck in video diffusion models:
>
> **1. Temporal Complementarity**: Based on diffusion denoising theory, the generation process naturally progresses from low to high-frequency content. ProRes leverages this by using lower resolutions during early timesteps when content structure is being established, while Attention Carving becomes most effective during later timesteps when high-resolution details are refined but full attention is no longer necessary.
>
> **2. Spatial Complementarity**: ProRes reduces the **total number of tokens** (addressing the N in O(N²) complexity), while Attention Carving focuses on reducing the **density of token interactions in high resolution** (addressing the practical computation within the O(N²) framework). This dual approach creates a multiplicative effect on acceleration.
>
> **3. Content-Generation Constraints and Resolution Trade-offs**: During content generation phases, AttenCarve cannot employ extremely high attention sparsity levels without compromising content quality, thus limiting its acceleration potential. However, content generation fundamentally does not require high-resolution inputs. ProRes strategically exploits this by reducing token interactions during content generation through lower resolutions, while preserving the ability to apply higher sparsity levels during high-resolution detail refinement when content structure is already established. This creates a **sparsity-resolution scheduling**: conservative sparsity at low resolution (content generation) and aggressive sparsity at high resolution (detail refinement), maximizing acceleration while maintaining generation quality.
>
> **4. Orthogonality**: Unlike competing acceleration methods that often interfere with each other, ProRes and Attention Carving operate on orthogonal dimensions--resolution scaling vs. attention sparsity--allowing them to be combined without negative interactions. This explains why our combined approach achieves 7.22× speedup compared to individual contributions of 3.28× (ProRes) and 2.17× (Attention Carving).
>
> ## Q6 Ablation on Mask Strategies
>
> In Table 3(b) and Figure 7, we provided an ablation study on SFC selection and mask selection. We acknowledge that more comprehensive ablation studies (i.e., incrementally adding strategies) would better validate our method's effectiveness. Here we present a more detailed ablation study to provide better validation of our method:
>
> |Mask Type|Importance|Condition|Adjacency|VBench|Latency|
> |-|-|-|-|-|-|
> |No Mask (FA)|Full|Full|Full|82.57%|495s|
> |TopK only|✓ (partially)|||81.35%|220s|
> |TopK, SFC|✓ (partially)|||81.41%|220s|
> |TopK, Prob. Constraint, SFC|✓|||81.87%|223s|
> |w/o Adjacency|✓|✓||82.42%|222s|
> |w/o Condition|✓||✓|81.82%|221s|
> |w/o Importance||✓|✓|77.41%|140s|
> |All Mask|✓|✓|✓|83.07%|225s|
>
> This comprehensive ablation demonstrates that: (1) Each mask component contributes meaningfully to both quality and efficiency, (2) The importance mask provides the most significant impact on both metrics, (3) The combination of all three masks achieves the optimal balance between generation quality and computational efficiency.
>
> For additional details regarding mask-related implementations, please refer to our response in **G2mf Q1.1**.

---

> > ### Comment · Reviewer_cBBb · 2025-08-09
> >
> > Thank you for your detailed response which has addressed most of my concerns. However, the explanations regarding the Title and Method in Q4 and Q5 seem not convince me. So I decided to maintain the current score.

---

> > > ### Author Response · Authors · 2025-08-09
> > > **Thank you!**
> > >
> > > Dear reviewer cBBb,
> > >
> > > Thank you for carefully considering our rebuttal and your continued positive perspective on this work. We understand your remaining reservations about the title and the relationship between the two components.
> > >
> > > **We summarized this as the core motivation stated in Q5(3)**:
> > >
> > > > During content generation phases, AttenCarve cannot employ extremely high attention sparsity levels without compromising content quality, thus limiting its acceleration potential. However, content generation fundamentally does not require high-resolution inputs. ProRes strategically exploits this by reducing token interactions during content generation through lower resolutions, while preserving the ability to apply higher sparsity levels during high-resolution detail refinement when content structure is already established.
> > >
> > > Our intent is a single “carving” framework: Progressive Resolution (ProRes) reduces token count early (acting on N), while Attention Carving sparsifies interactions later (acting on interaction density); they are temporally complementary and together yield a multiplicative speedup (7.22× vs. 3.28×/2.17× individually) with high-quality generation.
> > >
> > > To make this clearer, we will revise the introduction to explicitly present both components and update the method to show their staged coupling. We appreciate your time and constructive engagement.

---

### Official Review · Reviewer_iQ7G · 2025-06-19

**Clarity:** 2
**Significance:** 3
**Originality:** 2
**Rating:** 4
**Confidence:** 4

**Summary:**

The paper presents Jenga, a plug-and-play, training-free method to accelerate video diffusion Transformers, reducing inference time from minutes to seconds. Contributions include the introduction of block-wise attention carving for sparse computation, as well as the incorporation of progressive resolution for further acceleration. Additionally, Jenga is applied during inference without requiring any training.

**Questions:**

1.	Regarding novelty: It is recommended that the authors reframe the motivation and its correspondence with the proposed method to better highlight the novelty of the approach.
2.	Regarding acceleration performance: The authors are encouraged to provide more detailed explanations. If the acceleration is based on TeaCache, the speedup appears to be modest.
3.	Regarding metrics: It is suggested that the authors include objective metrics such as FVD to provide a more comprehensive evaluation.

**Ethical Concerns:**

["NO or VERY MINOR ethics concerns only"]

**Final Justification:**

The author's response resolved my confusion and I decided to maintain my rating.

**Limitations:**

yes

**Paper Formatting Concerns:**

The paper has no formatting issues.

**Quality:**

3

**Strengths And Weaknesses:**

Strengths:

1.	The paper has a well-structured format and the illustrations are visually appealing.
2.	The training-free design enhances its generalizability.
3.	The design of the masks in the method is quite clever and insightful.
4.	The experiments are conducted across multiple methods, demonstrating the generalizability of Jenga.

Overall, the design of the mask and the complete experiments make me inclined to accept this work.

---

Weakness:

1.	The observation that early denoising steps in diffusion do not require high-resolution latents is not novel and has been previously discussed in the literature. The progressive resolution design appears more heuristic than principled. Additionally, there seems to be a potential inconsistency between the claim in Lines 45–47 that early steps establish global structure, and the observation in Line 60 that low-resolution generation tends to emphasize local, zoomed-in details. Clarifying this relationship would strengthen the motivation for ProRes.
2.	The sparse attention mechanism proposed in Jenga shares similarities with prior work such as CLEAR [1] and SVG [2], which also adopt token or block sparsity. While the proposed block-wise dynamic selection is interesting, it may not represent a fundamentally new direction compared to these existing techniques.
3.	In Table 2(a), the acceleration improvement of Jenga over TeaCache for Wan2.1 appears relatively modest (from 34s to 24s). It would be helpful to clarify whether this result is achieved by applying Jenga on top of TeaCache or independently. This distinction is important for evaluating the true incremental benefit of the method.
4.	The evaluation currently focuses primarily on perceptual and semantic alignment metrics such as VBench and CLIPScore. However, the absence of structural fidelity metrics (e.g., FVD[3]) makes it difficult to assess the overall visual consistency and realism of the generated videos. Including such measures would provide a more comprehensive assessment of generation quality.

[1] Liu S, Tan Z, Wang X. CLEAR: Conv-Like Linearization Revs Pre-Trained Diffusion Transformers Up[J]. arXiv preprint arXiv:2412.16112, 2024.

[2] Xi H, Yang S, Zhao Y, et al. Sparse VideoGen: Accelerating Video Diffusion Transformers with Spatial-Temporal Sparsity[J]. arXiv preprint arXiv:2502.01776, 2025.

[3] Unterthiner T, Van Steenkiste S, Kurach K, et al. Towards accurate generative models of video: A new metric & challenges[J]. arXiv preprint arXiv:1812.01717, 2018.

---

> ### Author Rebuttal · Authors · 2025-07-28
>
> Dear reviewer iQ7G,
>
> Thank you for your valuable feedback and constructive comments. We are grateful for your positive assessment of our work, particularly your recognition of Jenga's generalizability and comprehensive experiments. We will address your comments below and incorporate the necessary revisions in the updated paper.
>
> ## Q1 The Novelty & Motivation of Jenga
>
> ### Q1.1 ProRes: Low-to-High Resolution
> We agree that multi-stage resolution progression is not entirely novel in isolation. However, several challenges arise in training-free pipelines, including the abnormal FOV problem discussed in L60. Jenga is the first training-free open-sourced progressive generation pipeline for Video Diffusion Transformers. As stated in **G2mf Q5**, most importantly, **ProRes solves the content degradation when adopting high sparsity attention across all timesteps, maintaining both efficiency and quality throughout the generation process.**
>
> **Regarding the inconsistent expression**, `early steps` and `generate low resolution contents` represent distinct concepts. In high-resolution generation, early steps produce global content while maintaining normal field of view (FOV). Similarly, in low-resolution generation, early steps also generate global content encompassing overall layout and structures. However, low-resolution content inherently introduces a "zoom-in" FOV degradation, where the generated layout and structures appear zoomed-in, presenting only partial views rather than the intended global perspective. We will clarify this distinction in our revision.
>
> ### Q1.2 AttenCarve: Sparse Attention is Not New
> We acknowledge that utilizing sparsity for attention acceleration is not fundamentally novel. Our work does not claim to be "the first sparse attention design for video generation." Instead, we focus on how to effectively leverage sparse attention through principled design choices and achieve higher sparsity levels while maintaining quality.
>
> Current widely-used attention approaches suffer from limitations: they are either static (lacking content adaptivity, **cBBb Q2**) or achieve only low sparsity levels ($\sim$0.5). Our method introduces SFC-based block partitioning and cutoff probability mechanisms to address resolution-dependent challenges and achieve substantially higher sparsity ($\sim$0.75-0.80) without quality degradation. The technical contribution lies in the systematic combination of these components to create a practical, training-free acceleration framework that works across different resolutions and model architectures.
>
> ### Q1.3 Complementary Nature of ProRes and Attention Carving
>
> The core innovation of Jenga lies in recognizing that **attention sparsity and progressive resolution are fundamentally complementary** techniques that address different aspects of the computational bottleneck in video diffusion models:
>
> **1. Temporal Complementarity**: Based on diffusion denoising theory, the generation process naturally progresses from low to high-frequency content. ProRes leverages this by using lower resolutions during early timesteps when content structure is being established, while Attention Carving becomes most effective during later timesteps when high-resolution details are refined but full attention is no longer necessary.
>
> **2. Spatial Complementarity**: ProRes reduces the **total number of tokens** (addressing the N in O(N²) complexity), while Attention Carving focuses on reducing the **density of token interactions in high resolution** (addressing the practical computation within the O(N²) framework). This dual approach creates a multiplicative effect on acceleration.
>
> **3. Content-Generation Constraints and Resolution Trade-offs**: During content generation phases, AttenCarve cannot employ extremely high attention sparsity levels without compromising content quality, thus limiting its acceleration potential. However, content generation fundamentally does not require high-resolution inputs. ProRes strategically exploits this by reducing token interactions during content generation through lower resolutions, while preserving the ability to apply higher sparsity levels during high-resolution detail refinement when content structure is already established. This creates a **sparsity-resolution scheduling**: conservative sparsity at low resolution (content generation) and aggressive sparsity at high resolution (detail refinement), maximizing acceleration while maintaining generation quality.
>
> **4. Orthogonality**: Unlike competing acceleration methods that often interfere with each other, ProRes and Attention Carving operate on orthogonal dimensions--resolution scaling vs. attention sparsity--allowing them to be combined without negative interactions. This explains why our combined approach achieves 7.22× speedup compared to individual contributions of 3.28× (ProRes) and 2.17× (Attention Carving).
>
> ## Q2 Structural Fidelity Metrics
>
> Thank you for raising this important point about evaluation metrics. We acknowledge that structural fidelity metrics like FVD (Fréchet Video Distance) provide additional validation of our method's effectiveness. While our primary evaluation focuses on VBench for comprehensive video quality assessment, we recognize the value of complementary metrics that capture different aspects of video quality. We also provide detailed VBench-Q quality assessments in **Appendix Table 6**, including video consistency, visual quality, smoothness, dynamic degree, etc.
>
> We conducted additional experiments with FVD to provide a more complete evaluation. As FVD evaluates the distance between **training data distribution** and **inference data distribution**, as a training-free method, we adopt 1. the **official** VBench test result (with different sampling seeds) and 2. the high-quality video dataset Inter4K as real data distributions to evaluate FVD.
>
> **The tables in Q3 further compare methods using FVD metrics**. The results show that Jenga preserves video generation quality comparable to the baseline, achieving an FVD score of 141 versus 144 in terms of structural fidelity. The FVD results align with the VBench trends observed across different methods.
>
> ## Q3 TeaCache & Jenga: Independent Results (with FVD metrics)
>
> We appreciate your interest in understanding the relationship between our work and complementary acceleration techniques like TeaCache. The experiments with TeaCache demonstrate that Jenga achieves substantial acceleration independently and can be combined with other orthogonal methods, including feature reuse and few-step models (**G2mf Q3**).
>
> |Method|VBench|Latency|Speed-Up|FVD-Inter4K ↓|FVD-Hunyuan ↓|
> |-|-|-|-|-|-|
> |HunyuanVideo|82.74%|1625s|1.00×|600|144|
> |AttenCarve Only|83.42%|748s|2.17×|448|164|
> |ProRes Only|82.85%|1075s|1.51×|620|191|
> |ProRes + Skip|82.57%|495s|3.28×|630|203|
> |AttenCarve + ProRes + Skip (Jenga-Turbo)|83.07%|225s|7.22×|583|141|
> |**AttenCarve + ProRes**|83.12%|485s|3.35×|542|127|
>
> Here we also report updated performance on the Wan2.1 model. Due to time constraints, we only tested AttenCarve + Skip on the Wan series in the submission. We have updated the Jenga-Turbo setting, which further boosts efficiency.
>
> |Method|VBench|Latency|Speed-Up|FVD-Inter4K ↓|FVD-Wan2.1 ↓|
> |-|-|-|-|-|-|
> |Wan2.1-1.3B|83.28%|115s|1.00×|788|182|
> |TeaCache-fast|82.63%|34s|3.48×|738|232|
> |AttenCarve + Skip (Jenga-Base)|82.68%|24s|4.79×|702|184|
> |AttenCarve Only|82.96%|71s|1.62×|674|169|
> |AttenCarve + ProRes + Skip (Jenga-Turbo)|82.52%|17s|**6.52×**|548|218|
> |**AttenCarve + ProRes**|82.86%|52s|2.21×|565|214|
>
> As discussed in **Q2 and G2mf Q3**, our results demonstrate that Jenga's acceleration is fundamentally independent of feature caching techniques like TeaCache. **We believe that further improvement on the basis of a mature acceleration method is both challenging and necessary.** One reason for the less significant speedup ratio is that the native input resolution (480P videos) is relatively small, resulting in a smaller proportion of attention computation. However, the reduction in the timestep dimension is linear. While TeaCache focuses on reusing computed features across timesteps, Jenga reduces computation through progressive resolution and selective attention, while also boosting generation quality. This orthogonality means the methods can be combined for potentially even greater acceleration benefits without quality degradation.
>
> **Future Integration:** We plan to explore combinations with TeaCache and other orthogonal efficient generation methods in future work, such as token purning, SageAttention (quantized attention), etc.
>
> [1] Stergiou, Alexandros and Poppe, Ronald, AdaPool: Exponential Adaptive Pooling for Information-Retaining Downsampling, 2021

---

> > ### Comment · Reviewer_iQ7G · 2025-08-03
> >
> > Q1: In the first table, FVD (the last row) surpasses the original performance, but in the second table, it does not. Why?
> > Q2: In Table 2, the acceleration effect of the last row is obviously not as good as teacache.

---

> ### Author Response · Authors · 2025-08-04
> **[Response for Q1] Thank you for your comments!**
>
> Dear reviewer iQ7G
>
> Thank you for your timely comments! We will address each question below:
>
> ## Q1. In the first table, FVD (the last row) surpasses the original performance, but in the second table, it does not. Why?
>
> Your observation is very astute! FVD metrics are related to the video generation distribution itself, particularly the potential changes in generated content domain when introducing ProRes. **As we can see, after introducing ProRes, the FVD-OriginModel results become "worse", but the FVD-Inter4K results become "slightly better".** This indicates that while our method may have some distribution differences from the original model (even though these differences, as explained in A1, fall within normal fluctuations), these differences may be related to the **model architecture itself**.
>
> 1. Since HunyuanVideo uses the mmDiT architecture while Wan uses the Self-Cross Attention architecture, the attention score amplifier we designed to solve FOV degradation in mmDiT cannot be fully utilized in Self-Cross Attention. Even if we simply increase the cross-attention weights, we cannot directly modify the self-attention results through this weight. Therefore, there may be some differences in results, but we believe these differences are acceptable. We must also acknowledge that some designs are indeed more suitable for the mmDiT architecture.
>
> 2. **This difference does not affect the distribution gap between our method and high-quality real-world videos (FVD-Inter4K)**, ensuring that the generated videos maintain high quality. As mentioned in our response, VBench includes multiple metrics related to video quality. We provide detailed VBench-Q quality assessments in Appendix Table 6, including video consistency, visual quality, smoothness, dynamic degree, etc.
>
> 3. FVD metric reliability: We contend that FVD metrics inherently allow for a certain degree of fluctuation. When we conducted inference using a different seed, the results improved substantially. We believe that metrics within reasonable ranges do not necessarily indicate degradation. **Moreover, removing TeaCache from Jenga-Turbo consistently outperforms TeaCache-Only across all quality metrics.**
>
> |Method|VBench|Latency|Speed-Up|FVD-Inter4K ↓|FVD-Wan2.1 ↓|
> |-|-|-|-|-|-|
> |Wan2.1-1.3B|83.28%|115s|1.00×|788|182|
> |TeaCache-fast|82.63%|34s|3.48×|738|232|
> |[Replace Seed] **AttenCarve + ProRes**|82.86%→83.56%|52s|2.21×|565→579|**214→189**|
> |[Replace Seed] AttenCarve + ProRes + Skip (Jenga-Turbo)|82.52%→82.80%|17s|**6.52×**|548→534|**218→194**|

---

> ### Author Response · Authors · 2025-08-04
> **[Response for Q2] Thank you for your comments!**
>
> ## Q2. In Table 2, the acceleration effect of the last row is obviously not as good as TeaCache.
>
> The differences between methods across the two tables have been mentioned in the original Q3.
> > One reason for the less significant speedup ratio is that the **native input resolution (480P videos) is relatively small, resulting in a smaller proportion of attention computation.**
>
> HunyuanVideo's input is 115K tokens total, while Wan2.1-1.3B's input is 31K tokens total. This causes the attention computation time to account for a much smaller proportion of the overall computation (77.8% for 115K and 47.1% for 31K), which is precisely the core acceleration principle of Jenga. Additionally, due to different attention sparsity levels between the two models, we set different cutoff probabilities $p$, which also affect the acceleration ratio of corresponding operators.
>
> Meanwhile, TeaCache's setting proportions differ between the two models. **In HunyuanVideo, TeaCache runs for 23/50 inference steps, while in the Wan2.1 series, TeaCache only runs for 15/50 steps.** Below is a simple revisit of the tables in the rebuttal:
>
> |Method|VBench|TokenNum|Latency|Speed-Up|FVD-Inter4K ↓|FVD-Origin ↓|
> |-|-|-|-|-|-|-|
> |HunyuanVideo|82.74%|115K|1625s|1.00×|600|144|
> |+TeaCache|82.39%|115K|703s|2.31×|648|156|
> |+ **AttenCarve + ProRes**|83.12%|115K|485s|3.35×|542|127|
> |+ AttenCarve + ProRes + Skip (Jenga-Turbo)|83.07%|115K|225s|7.22×|583|141|
> ||
> |Wan2.1-1.3B|83.28%|31K|115s|1.00×|788|182|
> |+TeaCache|82.63%|31K|34s|3.48×|738|232|
> |+ **AttenCarve + ProRes**|82.86%|31K|52s|2.21×|565|214|
> |+AttenCarve + ProRes + Skip (Jenga-Turbo)|82.52%|31K|17s|6.52×|548|218|
>
> We respectfully request that you acknowledge and understand our research efforts. **Our method focuses on "token carving," which reduces token interactions, so it is fundamentally different from feature-reuse at the methodological level.** Even with very few token inputs or low attention sparsity, TeaCache may be faster than our proposed method. We did not simply comparing AttenCarve + ProRes with TeaCache, as they essentially address problems from two orthogonal dimensions. While our method demonstrates greater utility for high-resolution and large-scale models, **we position it not as a direct competitor to TeaCache, but rather as a complementary approach addressing different optimization dimensions.**
>
> Most importantly, we can achieve additional acceleration on top of methods like TeaCache without performance degradation. However, TeaCache cannot be used together with few-step distilled models, such as AccVideo (5NFE, Jenga achieved a 3.16× speed-up, discussed in G2mf Q3).
>
> Therefore, we are not concerned that using AttenCarve + ProRes achieves lower acceleration ratios than TeaCache in certain specific scenarios, because **the combined results of the Jenga and TeaCache are better than using TeaCache alone in terms of both efficiency and quality**.
>
> **We will include the above discussion in our revision and emphasize the importance and orthogonality of the timestep skip.** Thank you very much for your insights and response!

---

> > ### Comment · Reviewer_iQ7G · 2025-08-06
> >
> > The author's response resolved my confusion and I decided to maintain my rating.

---

> > > ### Author Response · Authors · 2025-08-06
> > > **Thank you!**
> > >
> > > Dear reviewer iQ7G,
> > >
> > > Thank you again for your valuable comments and your continued positive perspective on this work! In the revision, we will include **the clarification in Q1, the FVD metric in Q2, and the importance and orthogonality of the timestep skip in Q3**

---

### Official Review · Reviewer_N3G7 · 2025-06-23

**Clarity:** 3
**Significance:** 4
**Originality:** 3
**Rating:** 5
**Confidence:** 5

**Summary:**

This paper proposes a novel training-free and efficient method for video generation. The approach is based on two key insights: (1) early denoising steps do not require high-resolution latents; and (2) later steps do not require dense attention. Based on these observations, the authors introduce a block-wise attention mechanism using 3D space-filling curves, which effectively reduces the number of key-value blocks in attention. Additionally, they present a multi-stage Progressive Resolution strategy that uses varying latent resolutions across different diffusion timesteps, thereby reducing token interactions throughout the process. By integrating a few-step approach, the method achieves 4–8× acceleration on HunyuanVideo-T2V without compromising video quality.

**Questions:**

See the weaknesses.

**Ethical Concerns:**

["NO or VERY MINOR ethics concerns only"]

**Final Justification:**

This paper accelerates the inference speed of VDM by incorporating sparse attention and progressive resolution denoising. Compared to existing sparse attention strategies, the proposed method achieves higher sparsity while maintaining generation quality. Moreover, when combined with current approaches, it achieves an 8× speed-up. Therefore, I believe this paper meets the standard for acceptance.

**Limitations:**

Yes.

**Paper Formatting Concerns:**

No major formatting issues.

**Quality:**

4

**Strengths And Weaknesses:**

Strengths:
1. Compared to prior methods with hand-crafted attention strategies, the proposed dynamic Token Carving mechanism more effectively identifies crucial key-value pairs in video content. It also preserves global context through accumulated probability.

2. The method maintains video generation quality while significantly improving inference efficiency.

3. Comprehensive experiments are conducted to validate the effectiveness of the proposed approach.

Weaknesses:
1. How are the diffusion timesteps for different stages determined in the Progressive Resolution strategy? What impact do different transition points have on the final video quality?

2. In line 287, the paper states that "a large cutoff value (p = 0.5) disrupts the selection balance among attention heads, leading to slight performance degradation." Could the authors elaborate on the mechanism behind this observation? A more detailed explanation or visualization would be helpful.

3. What is the speedup achieved when using only AttenCarve and ProRes (without applying timeskip)?

---

> ### Author Rebuttal · Authors · 2025-07-28
>
> Dear reviewer N3G7,
>
> Thank you for your valuable feedback and constructive comments. We are grateful for your **clear positive** assessment of our work, particularly your recognition of its dynamic token carving strategy and comprehensive experiments. We will address your comments below and incorporate the necessary revisions in the updated paper.
>
> ## Q1 Stage Ablation for Progressive Resolution
>
> Thank you for this insightful question. This is indeed an important ablation study we conducted but did not systematically report. We provide supplementary experiments in the table below:
>
> |Low-Res Step|10%|30%|50%|70%|90%|
> |-|-|-|-|-|-|
> |Traj-Timestep|987|947|883|767|437|
> |VBench|82.36%|82.35%|83.07%|81.03%|78.74%|
> |Latency|286s|253s|225s|207s|169s|
> * The experiment uses Jenga-Turbo's basic setting, containing one low-res > high-res stage transition.
>
> We have included a detailed analysis in **G2mf Q2** as follows:
> >The experimental results reveal key insights about trajectory-guided resolution scheduling. Due to the timestep shift mechanism in DiT, more computational steps are allocated to the early stages of the trajectory. Even in the final 10% of timesteps, the actual trajectory still contains approximately 43.7% of the total steps.
>
> >We observe a dramatic quality degradation when low-res steps exceed 50% of the total timesteps, while **introducing latent misalignment artifacts**, confirming that multi-stage generation requires sufficient high-resolution denoising to maintain video quality. Meanwhile, when stage 1 (low-res) steps are too few, quality also decreases. This aligns with our motivation (L48-49) that low-res content generation requires a higher attention density to establish proper content structure. This finding validates why ProRes and AttenCarve work synergistically--they are not merely independent methods but complementary components that address different phases of the generation process.
>
> >The 50% low-res step configuration achieves the best balance between quality (83.07%) and efficiency (225s), demonstrating that trajectory curvature analysis effectively guides optimal resolution scheduling decisions.
>
> Meanwhile, ensuring sufficient high-resolution steps is indeed a critical factor for maintaining generation quality in Jenga. This is partially a consequence of adhering to our training-free methodology. Some works like FlashVideo propose that "high resolution generation is relatively easy to optimize into few steps." Training a few-step detail generation model is certainly possible; however, such approaches require substantial data and computational resources, and may be superseded by stronger foundation generation models. Our training-free approach provides a more sustainable and generalizable solution that can readily adapt to newer and more powerful foundation video generation models without requiring retraining.
>
> ## Q2 The Impact of Cutoff Probability
>
> This is a valuable question. The cutoff probability threshold $p$ is one of the key hyperparameters that controls the sparsity-quality trade-off in our attention carving mechanism.
>
> We conducted systematic experiments to understand its impact in HunyuanVideo, with a Jenga-Turbo setting.
>
> |Cutoff Probability $p$|0.0|0.3|0.5|0.7|0.9|1.0 (Full Attention)|
> |-|-|-|-|-|-|-|
> |VBench|82.85%|83.07%|82.75%|82.18%|82.59%|82.57%|
> |Latency|227s|225s|232s|271s|330s|495s|
> |Effective Sparsity|80.3%|80.1%|78.5%|72.3%|57.5%|0.0%
> |STD Across Heads|0|545|19001|23546|70551|0|
> |Diff with Full Attention|0.141|0.136|0.132|0.115|0.074|0|
> * The latency results may have some fluctuations (not exceeding 5s), but the overall trend is correct.
>
> From these results, we derive the following analysis: Higher $p$ values make the model results approach Full Attention. We computed the standard deviation (STD) of selected block quantities across different heads and the average difference between AttenCarve and full attention. Due to varying feature extraction capabilities of different heads (visualized in Appendix C.2), the STD increases with higher $p$ values. We also observe that the local nature of attention is revealed: Effective Sparsity is significantly greater than the theoretical average selection value (57.5% vs. 10%).
>
> We believe that both low and high cutoff probability values are reasonable in this context: With high $p=0.9$, covering more probability regions ensures a certain degree of sparsity while allowing each head to retain most of the original features. Although $p=0.9$ does not achieve optimal performance in HunyuanVideo, it closely approaches the full attention baseline (82.57%, same as ProRes + timeskip in **Table 1**) and achieves the best results in Wan2.1.
>
> With lower $p=0.3$, AttenCarve selects the most important blocks, with the removal of long-tailed features in attention **balancing the block selections among heads (with a small STD)** and boosting performance. We guarantee higher sparsity while partially preserving the most important components in certain global heads.
>
> Intermediate $p$ values (0.5, 0.7) introduce relatively unimportant and partial features in attention heads (evidenced by the dramatic STD increase across heads without fully preserving global head representations), thereby reducing sparsity without improving results. We will incorporate this discussion into the revision, and thank you for prompting us to explore these ablation phenomena more deeply.
>
> ## Q3 Module-Wise Speed-Up Ablations
>
> We provided individual results for AttenCarve and ProRes in Table 1. AttenCarve and ProRes achieve speedups of 2.17× and 1.51× respectively (3.28× when combined with timestep skip). For a module-wise evaluation, we provide detailed parameter configurations in Table 4 in the Appendix. Below is an excerpt from Table 1, with additional module combinations. We use a two-stage basic setting (Jenga-Turbo), where attention sparsity is also applied in a two-stage manner in AttenCarve experiments.
>
> |Method|VBench-Q|VBench-S|VBench|Latency|Speed-Up|
> |-|-|-|-|-|-|
> |Baseline|85.21%|72.84%|82.74%|1625s|1.00×|
> |AttenCarve Only|85.31%|75.85%|83.42%|748s|2.17×|
> |ProRes Only|86.20%|69.43%|82.85%|1075s|1.51×|
> |ProRes + Step Skip|85.78%|69.73%|82.57%|495s|3.28×|
> |AttenCarve + ProRes|84.65%|76.98%|83.12%|485s|3.35×|
> |Jenga-Turbo (Combined)|84.47%|77.48%|83.07%|225s|7.22×|
>
> The results demonstrate that **the acceleration ratio can essentially be viewed as the product of individual method speedups**. The synergistic combination of these approaches enables Jenga to achieve high acceleration ratios while maintaining video generation quality. The organic integration of ProRes and AttenCarve working together produces even greater performance gains than the sum of their individual contributions would suggest.

---

> > ### Comment · Reviewer_N3G7 · 2025-08-04
> >
> > In Q2, the authors computed STD of selected block quantities. I would like to understand how the STD was calculated and what it represents.

---

> ### Author Response · Authors · 2025-08-04
> **Thank you for your comments!**
>
> Dear reviewer N3G7,
>
> Thank you for your timely comments! We will address your question below:
>
> ## Q1. STD of selected block quantities
>
> We apologize for not providing a detailed implementation of the STD calculation. We used 500 prompts and recorded the attention head block selection values for each attention forward pass, then averaged these values. The specific formula is as follows:
>
> For the importance mask $\mathbf{B}^{h}_{\text{top}} \in \mathbb{R}^{M \times M}$ (where $h$ denotes the attention head and $M$ is the number of blocks, $M=900$ for one 720P HunyuanVideo inference), we compute the standard deviation across attention heads:
>
> $$\text{STD} = \sqrt{\frac{1}{H} \sum_{h=1}^{H} ||\mathbf{B}^{h}_{\text{top}} - \mu||_F^2}$$
>
> where $\mu = \frac{1}{H} \sum_{h=1}^{H} \mathbf{B}^{h}_{\text{top}}$ is the mean block selection mask across all heads, and $||\cdot||_F$ denotes the Frobenius norm.
>
> The final reported STD is averaged across all timesteps $T$, layers $L$, and the number of sampled prompts throughout the generation process.
>
> This metric measures the diversity and consistency of different attention heads in their block selection strategies. **Thus, it serves as a reference for block selection balance.**

---

> > ### Comment · Reviewer_N3G7 · 2025-08-05
> >
> > The author's response addressed all my concerns, so I decided to maintain the current score.

---

> > > ### Author Response · Authors · 2025-08-06
> > > **Thank you!**
> > >
> > > Dear reviewer N3G7,
> > >
> > > Thank you again for your valuable comments and your continued positive perspective on Jenga! We will include the corresponding **ablation studies and analyses for Q1, Q2, and Q3 in the revision.**

---

### Official Review · Reviewer_G2mf · 2025-07-02

**Clarity:** 3
**Significance:** 3
**Originality:** 3
**Rating:** 4
**Confidence:** 4

**Summary:**

This paper proposes Jenga, a training-free inference acceleration framework for video diffusion transformers (DiTs). It addresses two primary bottlenecks in DiT-based video generation: the quadratic complexity of self-attention and the multi-step nature of diffusion. Jenga achieves acceleration through two core components:

1. Attention Carving: A dynamic block-wise sparse attention mechanism using 3D space-filling curves to reduce redundant token interactions.

2. Progressive Resolution (ProRes): A multi-stage generation strategy where early denoising occurs at low resolutions and progressively upsamples latents, reducing the total number of processed tokens.

The framework is shown to be broadly applicable across text-to-video, image-to-video, and distilled models, achieving up to 8.83× speedup on state-of-the-art video diffusion models like HunyuanVideo, AccVideo, and CogVideoX, with minimal degradation in generation quality (e.g., less than 0.01% drop on VBench).

**Questions:**

1. Clarification of Block-Level Attention Mechanism: In Figure 2 and Section 3.1, it's unclear how frequently the block-wise attention mask is updated. More clarification is needed on the roles and update frequencies of the importance mask, condition mask, and adjacency mask—specifically, which are static and which contribute most to performance. Additionally, please elaborate on why the condition and adjacency masks are solely determined by resolution.

2. Connection Between Trajectory Curvature and Resolution Scheduling: The paper introduces curvature analysis to guide block selection, while the progressive resolution strategy (ProRes) appears independently designed. Is there any correlation between curvature changes in the sampling trajectory and the decision points for resolution upsampling?

3. Applicability to Low-Step or Consistency Models: Has Jenga been evaluated in low-step regimes (e.g., 4–8 denoising steps) or on consistency-based models like SD3? These increasingly popular settings may challenge assumptions in both ProRes and the sparsity design of Attention Carving.

4. Adaptivity of Sparsity Masks: The current block selection uses fixed thresholds (e.g., top-k, softmax cutoff). Could adaptive or learned sparsity mechanisms further improve the balance between quality and efficiency, especially across diverse prompt distributions or content styles?

**Ethical Concerns:**

["NO or VERY MINOR ethics concerns only"]

**Final Justification:**

After reading the authors' rebuttal and other reviews, I choose to maintain my recommendation.

**Limitations:**

The paper does address some limitations, especially regarding resolution transitions and heuristic design, but the discussion could be more thorough. In particular:

1. Lack of Generalization to Consistency and Low-Step Models: The current method assumes a moderately long sampling process (e.g., 24–50 steps). It may not generalize to ultra-fast samplers like consistency models or low-step DMs, where resolution transitions and sparsity patterns can break down.

2. Static Heuristics for Sparsity: The block selection (top-k, threshold-based) is fixed per stage. While effective, this limits adaptability to different prompts or domains and may underperform in highly dynamic content.

**Quality:**

3

**Strengths And Weaknesses:**

## Strengths
1. Proposes a training-free, plug-and-play framework for efficient video generation that achieves up to 8.83× speedup across various DiT architectures, including HunyuanVideo and AccVideo, without model modification or retraining.

2. Introduces Attention Carving, a dynamic, block-wise sparse attention mechanism guided by 3D space-filling curves and masks based on block importance, adjacency, and condition relevance—offering a clean and GPU-friendly design.

3. Leverages a Progressive Resolution (ProRes) strategy that denoises at lower resolutions in earlier stages and progressively increases resolution, aligning with the coarse-to-fine nature of diffusion processes.

4. Includes a novel text-attention amplifier to stabilize the field of view (FOV) during early low-resolution steps, which would otherwise generate zoomed-in or truncated content.

5.  strong empirical results across text-to-video, image-to-video, and distilled models, maintaining semantic quality and CLIP scores even under aggressive speedup (e.g., Jenga-Flash, 8.83×).

## Weaknesses

1. While the paper compares against cache-based methods, Jenga does not reuse intermediate states or features, which limits its benefit for very low-step (e.g., <8) samplers or consistency models, where reuse is more effective.

2. The block-wise attention masking procedure, though efficient, is still heuristic-based and uses static top-k and probability thresholds. No adaptive or learned sparsification is explored.

3. he motivation for combining curvature estimation with ProRes is underdeveloped: how the geometry of the sampling trajectory informs resolution transitions isn’t fully explained or connected.

4. The novelty is somewhat incremental, as both sparse attention (e.g., SpargeAttn) and progressive resolution (e.g., Bottleneck Sampling) have prior art. Jenga’s contribution lies more in effective integration than in introducing new algorithmic ideas.

---

> ### Author Rebuttal · Authors · 2025-07-28
>
> Dear reviewer G2mf,
>
> Thank you for your feedback and comments. We appreciate the thoroughness with which you have reviewed our manuscript, as evidenced by your insightful and valuable comments. We are grateful for your positive assessment of our work, particularly your recognition of its novelty and results. We will address your comments below and incorporate the necessary revisions in the updated paper.
>
> ## Q1 Clarification of Block-Level Attention
> We address your concerns by breaking them down into the following specific points:
>
> ### Q1.1 How frequently the block-wise attention mask is updated?
>
> The table below provides detailed information about the number of updates and computation timing for each mask type in an $L$-layer DiT model with $T$ actual denoising timesteps across $S$ resolution stages in ProRes.
>
> |Mask Type|Importance|Condition|Adjacency|
> |-|-|-|-|
> |Number of updates|$LT$|$S$, switch each stage|$S$, switch each stage|
> |When to compute|Start of attention| Before denoise |Before denoise|
>
> Attention Carving directly replaces the normal Full Attention. This means the head-block-wise attention mask $\mathbf{B}$ is updated in **per-layer attention**. If we perform inference on a 50-step model with 60 layers, it will update 50×60=3000 times. This design choice is motivated by the fact that attention patterns may vary significantly across different heads, layers and timesteps, as visualized in **Appendix Fig. 13**. We have experimented with reusing masks across different timesteps, but this approach failed to generate satisfying video results (VBench < 80%). We will include this explanation in the revision for better clarification. Additionally, while the block-selection operation (determining the importance mask $\mathbf{B}_\text{top}$) is performed layer-wise, we demonstrate in Appendix C.4 and Fig. 15 that **the computational overhead of block selection is minimal** (less than 3% of the total sparse attention computation).
>
> Regarding the Condition Mask and Adjacency Mask, as stated in L162-163,
>
> > In Jenga, Condition Mask and Adjacency Mask are pre-computed, and only determined by the resolution and the partition function $\mathcal{G}$.
>
> These two masks are static and precomputed as the resolution-aware SFC ($\mathcal{G}$) is predetermined. The only updates occur during stage switches when the latent resolution changes.
>
> ### Q1.2 Which Mask Contributes Most to Performance?
> In Table 3(b) and Figure 7, we provided an ablation study on SFC selection and mask selection, demonstrating the effectiveness of our mask selection scheme. Here we present a more comprehensive ablation study to provide better validation of our method. The "No Mask" baseline is the result with ProRes and timestep skip.
> |Mask Type|Importance|Condition|Adjacency|VBench|Latency|
> |-|-|-|-|-|-|
> |No Mask (FA)|Full|Full|Full|82.57%|495s|
> |TopK only|✓ (partially)|||81.35%|220s|
> |TopK, SFC|✓ (partially)|||81.41%|220s|
> |TopK, Prob. Constraint, SFC|✓|||81.87%|223s|
> |w/o Adjacency|✓|✓||82.42%|222s|
> |w/o Condition|✓||✓|81.82%|221s|
> |w/o Importance||✓|✓|77.41%|140s|
> |All Mask|✓|✓|✓|83.07%|225s|
>
> The above results demonstrate that: (1) Each mask component contributes meaningfully to quality or efficiency, (2) **The importance mask provides the most significant impact on both metrics**, (3) The combination of all three masks achieves the optimal balance between visual quality and computational efficiency. The "w/o Importance" case shows the most dramatic speedup but significant quality degradation, confirming that **content-aware adaptive block selection** is crucial for maintaining generation quality.
>
> ## Q2 Connection Between Trajectory Curvature and Resolution Scheduling
>
> The denoising trajectory serves as a guidance mechanism for optimal resolution scheduling. We conducted an ablation study on resolution scheduling to better understand the relationship between trajectory curvature and optimal transition timing:
> |Low-Res Step|10%|30%|50%|70%|90%|
> |-|-|-|-|-|-|
> |Traj-Timestep|987|947|883|767|437|
> |VBench|82.36%|82.35%|83.07%|81.03%|78.74%|
> |Latency|286s|253s|225s|207s|169s|
> * The experiment uses Jenga-Turbo's basic setting, containing one low-res > high-res stage transition.
>
> The experimental results reveal key insights about trajectory-guided resolution scheduling. Due to the timestep shift mechanism in DiT, more computational steps are allocated to the early stages of the trajectory. Even in the final 10% of timesteps, the actual trajectory still contains approximately 43.7% of the total steps.
>
> We observe a dramatic quality degradation when low-res steps exceed 50% of the total timesteps, while **introducing latent misalignment artifacts**, confirming that multi-stage generation requires sufficient high-resolution denoising to maintain video quality. Meanwhile, when stage 1 (low-res) steps are too few, quality also decreases. This aligns with our motivation (L48-49) that low-res content generation requires a higher attention density to establish proper content structure. This finding validates why ProRes and AttenCarve work synergistically--they are not merely independent methods but complementary components that address different phases of the generation process.
>
> The 50% low-res step configuration achieves the best balance between quality (83.07%) and efficiency (225s), demonstrating that trajectory curvature analysis effectively guides optimal resolution scheduling decisions.
>
> ## Q3 Discussion on Consistency and Low-Step Models
> We may not have clearly explained that **AccVideo is actually a low-step model (5 NFE)** derived from HunyuanVideo. In Table 2(b), we adopt Jenga on AccVideo with single and multi-GPU integration, achieving a 2.12× acceleration with only AttenCarve with comparable VBench and CLIPScore performance. Our method emphasizes the orthogonality with step-reduction (distilled model like AccVideo) and feature reuse (TeaCache) without affecting generation quality.
> |Method|VBench|CLIP|Latency|Speed-Up|
> |-|-|-|-|-|
> |AccVideo|83.82%|31.23|161s|1.00×|
> |AccVideo + AttenCarve (Paper Reported)|83.39%|31.07|76s|2.12×|
> |AccVideo + AttenCarve + ProRes|83.29%|31.12|51s|3.16×|
>
> Meanwhile, based on our discussion in Q2 regarding ProRes and trajectory analysis, as resolution switching fundamentally depends only on the corresponding trajectory timestep, we believe **few-step diffusion models can directly apply ProRes for further acceleration**. The table above demonstrates that using a two stage ProRes can maintain quality while providing 3.16× speedup even in few-step scenarios. This reinforces the orthogonal nature of our approach--it can complement existing few-step models without interference.
>
> ## Q4 Adaptivity of Sparsity Masks
>
> As discussed in Q1, the importance mask is updated adaptively in each attention forward pass, with adaptation for each attention head in our implementation. This means the block selection operation is online and adaptive.
>
> Although SFC is pre-determined by resolution, it has several benefits (a) adaptive resolution capability and (b) inherent local nature (which motivates our "Jenga" analogy). For a more detailed discussion on static or adaptive block partition, please refer to response in **cBBb Q2**.
>
> Although $k$ and threshold $p$ are pre-defined, the block selection for each query block remains adaptive across different attention heads based on the current content. As Jenga achieved a higher Dynamic Degree score in VBench in **Appendix Table 6**, we may not directly treat the high sparsity to the degradation in generating highly dynamic contents.
>
> As a pure training-free method, current version avoids the need for offline per-head optimal search procedures (a fitted static sparsification, but not adaptive with input feature) that methods like STA[1] require. Jenga's significantly smaller search space makes it easily adaptable to new models without extensive hyperparameter tuning. It would be a future direction to design a metric to determine adaptive or learnable sparsity.
>
> ## Q5 Novelty is Incremental?
>
> We acknowledge that there are previous works in sparse attention and multi-resolution sampling and discussed them in L92-93, L98-101. However, our contribution represents an advance beyond existing approaches:
>
> **Fundamental Motivation Difference:** Jenga's core motivation differs fundamentally from previous work. We treat resolution scaling and attention carving as a unified framework, both targeting **token interaction reduction**. The central challenge is balancing large attention sparsity with content generation quality. Our unified approach addresses this systematically.
>
> **Comparison with SparseAttn:** SparseAttn achieves only moderate sparsity (0.5) with extra tuning, compared to Jenga's high sparsity (0.75). This substantial improvement in sparsity-quality trade-off represents a meaningful technical contribution. AttenCarve's higher sparsity is achieved by the introduction of ProRes, which pre-generates (noisy) content in a low-res space.
>
> **Comparison with Bottleneck Sampling:** The BottleNeck design requires additional computation in the first stage and achieves only sub-optimal speedup ratios, even with some timestep reduction. Our goal is to achieve extreme speedup ratios through systematic optimization.
>
> **Technical Novelty:** Our method introduces several technical innovations: (1) 3D SFC-based block partitioning and adaptive block selection with hybrid masks, (2) Text-attention amplifier that addresses FOV problems in training-free low-res generation. **Most importantly, ProRes solves the content degradation when adopting high sparsity attention across all timesteps, maintaining both efficiency and quality throughout the generation process.** While technical directions may share similarities, we believe Jenga represents a technically solid contribution that combines strong practical utility in video generation.
>
> [1] P. Zhang .et.al., Fast Video Generation with Sliding Tile Attention, 2025

---

> > ### Comment · Reviewer_G2mf · 2025-08-07
> > **Reply**
> >
> > Thank you for your response and for addressing my concerns. I will keep my rating unchanged.

---

> > > ### Author Response · Authors · 2025-08-07
> > > **Thank you!**
> > >
> > > Dear reviewer G2mf,
> > >
> > > Thank you again for your valuable comments and your continued positive perspective on Jenga!
> > > In the revision, we will include the corresponding clarifications and ablations, **especially the updated result for few-step models in Q3, and ablations in Q1 and Q2**.

---

### Comment · Area_Chair_PSah · 2025-08-06
**Reminder: Author–Reviewer Discussion Closing Soon**

Dear Reviewers,

This is a gentle reminder that the Author–Reviewer Discussion phase ends within two days (by August 8). Please take a moment to read the authors’ rebuttal and engage in the discussion.

Your participation is important to ensure a fair and constructive review process. If you feel your concerns have been sufficiently addressed, you may also submit your Final Justification and update your rating early. Thank you for your contributions.

Best,

AC

---

### Note · Authors · 2025-08-12

We are deeply grateful to all reviewers for their positive assessments and the ACs' thorough engagement. **The consistent positive ratings across all reviewers, maintained after comprehensive discussions, reflect strong confidence in Jenga's technical merit and practical significance.** The review process has been invaluable in strengthening our work, and we deeply appreciate the time and expertise invested by each reviewer.

Jenga tackles one of the most pressing bottlenecks, which is making high-quality video generation practically deployable.
Reviewers consistently praised our "clever and insightful" design (iQ7G), "strong empirical results" (G2mf), and "comprehensive experiments" (N3G7, cBBb). Our extensive rebuttals provided detailed ablations and additional metrics, with reviewers confirming that most concerns were "addressed" and "resolved." The synergistic combination of Progressive Resolution and Attention Carving represents a principled approach to the O(N²) complexity challenge in Video Diffusion Transformers (DiTs).

As a training-free acceleration framework for Video Diffusion Transformers, Jenga offers immediate deployment across state-of-the-art models without retraining. Its orthogonality with existing models and compatibility with few-step models demonstrate its versatility for the rapidly evolving video generation research.

We believe this work represents the kind of practical, well-executed research that NeurIPS values—technically sound, immediately useful, and positioned to enable significant advances in the field. Thank you!

---

### Decision · Program_Chairs · 2025-09-17

**Decision:**

Accept (poster)

**Comment:**

This paper introduces Jenga, a tranining-free method for video DiTs which consists of attention carving and progressive resolution. Reviewers are satisfied with the novelty, performance and inference efficiency of the proposed method as well as the writing of the paper. Conerns raise by reviewers are addressed. In the end, the reviewers all recommend accept. After including the newly added important results as promised, I think this paper will meet the standard and therefore recommend an acceptance.